# Two fingerprinting sets for *Humulus lupulus* based on KASP and microsatellite markers

**Mandie Driskill**[1], **Katie Pardee**[1], **Kim E. Hummer**[1], **Jason D. Zurn**[2], **Keenan Amundsen**[3], **Annette Wiles**[4], **Claudia Wiedow**[5], **Josef Patzak**[6], **John A. Henning**[7], **Nahla V. Bassil**[1]*

**1** USDA-ARS, National Clonal Germplasm Repository, Corvallis, Oregon, United States of America, **2** Department of Plant Pathology, Kansas State University, Manhattan, Kansas, United States of America, **3** Department of Agronomy and Horticulture, University of Nebraska-Lincoln, Lincoln, Nebraska, United States of America, **4** Midwest Hops Producers, Plattsmouth, Nebraska, United States of America, **5** The New Zealand Institute for Plant and Food Research Limited, Palmerston North, New Zealand, **6** Hop Research Institute, Co, Ltd., Žatec, Czech Republic, **7** USDA-ARS, Forage Seed and Cereal Research Unit, Corvallis, Oregon, United States of America

\* Nahla.bassil@usda.gov

**Data Availability Statement:** All relevant data are within the manuscript, provided as links, and/or in its Supporting Information files. SSR genotype data for accessions from the NCGR are available in

## Abstract

Verification of clonal identity of hop (*Humulus lupulus* L.) cultivars within breeding programs and germplasm collections is vital to conserving genetic resources. Accurate and economic DNA-based tools are needed in dioecious hop to confirm identity and parentage, neither of which can be reliably determined from morphological observations. In this study, we developed two fingerprinting sets for hop: a 9-SSR fingerprinting set containing high-core repeats that can be run in a single PCR reaction and a kompetitive allele specific PCR (KASP) assay of 25 single nucleotide polymorphisms (SNPs). The SSR set contains a sex-linked primer pair, HI-AGA7, that was used to genotype 629 hop accessions from the US Department of Agriculture (USDA) National Clonal Germplasm Repository (NCGR), the USDA Forage Seed and Cereal Research (FSCR), and the University of Nebraska-Lincoln (UNL) collections. The SSR set identified unique genotypes except for 89 sets of synonymous samples. These synonyms included: cultivars with different designations, the same cultivars from different sources, heat-treated clones, and clonal variants. Population structure analysis clustered accessions into wild North American (WNA) and cultivated groups. Diversity was slightly higher in the cultivated samples due to larger sample size. Parentage and sib-ship analyses were used to identify true-to-type cultivars. The HI-AGA7 marker generated two male- and nine female-specific alleles among the cultivated and WNA samples. The SSR and KASP fingerprinting sets were compared in 190 samples consisting of cultivated and WNA accession for their ability to confirm identity and assess diversity and population structure. The SSR fingerprinting set distinguished cultivars, selections and WNA accessions while the KASP assays were unable to distinguish the WNA samples and had lower diversity estimates than the SSR set. Both fingerprinting sets are valuable tools for identity confirmation and parentage analysis in hop for different purposes. The 9-SSR assay is cost efficient when genotyping a small number of wild and cultivated hop samples (<96) while the KASP assay is easy to interpret and cost efficient for genotyping a large number of cultivated samples (multiples of 96).

GRIN-Global at https://npgsweb.ars-grin.gov/gringlobal/method?id=496582; KASP data for accessions from the NCGR are also available in GRIN-Global at https://npgsweb.ars-grin.gov/gringlobal/method?id=496583. Genotypic data for the 629 sample and R and Python scripts are available at the following github repository: github.com/ Bassil-Lab/(https://github.com/Bassil-Lab/Driskill_et_al_2021_PLOSONE_Humulus_two_fingerprint_sets ). TASSEL pipeline scripts are available at: https://sourceforge.net/p/tassel/tassel3-standalone/ci/master/tree/scripts/. SSR and KASP SNP genotypic data are available through GRIN Global at https://npgsweb.ars-grin.gov/gringlobal/method?id=496582 and https://npgsweb.ars-grin.gov/gringlobal/method?id=496583, respectively. KASP SNP names were given the "HK" prefix but reference and alternative alleles were removed from the ends of the names.

**Funding:** We thank USDA CRIS 2072-21000-049-00D and CRIS 2072-21000-051-00D for financial support for this project. We thank the Brewers Association for additional funding of this project. The funders had no role in study design, data collection and analysis, decision to publish, or preparation of the manuscript.

**Competing interests:** The authors have declared that no competing interests exist.

## Introduction

The commercial hop (*Humulus lupulus* L.) is a dioecious perennial plant of the Cannabaceae. This species has five botanical varieties, as determined by leaf morphology and geographic distribution [1,2]. Theses varieties are European hop (*H. lupulus* var. *lupulus* E. Small), Asian hop [*H. lupulus* var. *cordifolius* (Miq.) Maxim.], and three North American endemics hops (*H. lupulus* var. *lupuloides* E. Small, *H. lupulus* var. *neomexicanus* A. Nelson and Cockerell, and *H. lupulus* var. *pubescens* E. Small) [1,2]. Most cultivated hop plants are derived from the European hop, though other subspecies have contributed to the diversity of the breeding gene pool. Hop cultivars are mostly diploid ($2n = 2x = 20$), with nine autosomal chromosomal pairs and two sex chromosomes. Some cultivars are triploid and some breeding lines can be tetraploid [3,4].

The mature female inflorences, commonly called cones, are the economically important part of the hop plant. Production is in the United States, Europe, China, Australia, New Zealand, South Africa, and other regions [5,6]. The female cones contain lupulin glands, which produce chemical compounds that are desirable for flavoring beer and are used for aroma and bitterness [7,8]. Male plants are required for the breeding process but are excluded from cone production to reduce seed set, which results in lupulin contributing a higher proportion to hop cone weight [7,6]. In the United States, approximately 51 thousand tonnes of cones were harvested in 2019, totaling in an estimated value of $637 million [9]. The value of production in 2016 stood at $65 million—nearly doubling 2015's production value. This increase in production has been driven largely by an increase in the craft beer industry. In the last decade, the number of American craft breweries experienced a massive 298% increase in growth, with the number of tap rooms undergoing the greatest increase from 0 in 2010 to 3471 in 2020 [10]. The Barth-Haas Group [11] reported 146 hop cultivars in common use worldwide, which was an increase of 17 cultivars since 2009. To satisfy this need, farmers and brewers are establishing their own hop gardens with desired cultivars that they can use to impart the best aromas and flavors to their brew [12]. In the US, the number of craft breweries has doubled or more in all 50 states over the last decade [10]. To stay competitive, craft breweries are in search of hop cultivars with unique aromatic profiles to create new tastes for consumers beyond the classic citrus, floral, and herbal notes. The breeding programs at the US Department of Agriculture (USDA), Agricultural Research Services (ARS), including the Forage Seed and Cereal Research (FSCR), private companies, universities, and more recently at Oregon State University (OSU) are actively working to create new varieties for this burgeoning industry [13,14].

Hop cultivation began in Europe in the 1100s, where monasteries began producing beer and chose to incorporate hops for flavor with the accidental side benefit of antimicrobial activity for a beverage where drinking water was unclean [15]. Regional landrace selections of the wild hop from Saaz, Czech Republic, Backa, Serbia, and Hallertau and Tettnang, Germany came to be known for unique qualities and tastes. Hop breeding programs with long-term projects based on controlled crosses for plant improvement began after the 1900s [16,17]. Horticultural hop varieties were identified by geography or large morphological differences [1,2]. Hop plants were propagated by root cuttings, but the significance of maintaining specific clones of desirable types, i.e., "cloning," was not realized until the 20th century. Thus, plant performance, specifically yield, was highly variable because of variation in rainfall and pressure from diseases and pests. Farmers would realize a full yield perhaps once in a decade, leading to overplanting. This caused large fluctuations in the supply and price of hops between bumper and bust harvests.

In 1904, E. S. Salmon, a mycologist at Wye College (Southeastern Agricultural College), Kent, England, began a hop development program to address disease issues by applying new

principles of plant breeding. By 1917, Salmon had partnered with the East Malling Research Station to evaluate hop on a large scale [16]. Even then, Salmon used wild American material (*H. lupulus* var. *lupuloides)* from Manitoba, Canada to increase disease resistance and resin content and to improve aroma profiles [3,17]. From the Wye breeding program, Salmon produced three very successful cultivars: 'Brewer's Gold', 'Northern Brewer', and 'Bullion'. To date, most high-alpha or bitter breeding programs have used these three cultivars as parents [17]. Long-term breeding programs such as that at the USDA-ARS FSCR located in Corvallis, Oregon and Prosser, Washington, as well as newer programs at OSU and UNL are tasked with breeding locally adapted cultivars that are disease resistant to pathogens such as powdery and downy mildew and have increased yield and enhanced brewing characteristics. Modern breeding programs continue to utilize wild germplasm to develop new cultivars with different sources of disease resistance combined with new aroma and flavoring profiles [8,18]. Each of the aforementioned programs maintains germplasm collections containing experimental breeding material in addition to European cultivars and wild accessions.

Evaluating the diversity of germplasm is important to any hop breeding program to make informed crosses. In support of these breeding programs and to maintain historically important cultivars for posterity, the National Clonal Germplasm Repository (NCGR) in Corvallis, Oregon, manages a world hop collection with 647 accessions originating from 21 countries. The collection is highly diverse and contains European cultivars, regional landraces, and wild native North American accessions. At the NCGR, each clonal accession is propagated vegetatively and could easily be mis-identified if solely based on morphology [3], since morphology can change based on environmental influences such as growing conditions [8,19–21]. An economic DNA-based tool to confirm cultivar identity should be useful for management at the USDA and any other germplasm collection.

Two of the most common codominant DNA-based markers used for germplasm identification are microsatellites, also known as simple sequence repeats (SSRs), and single nucleotide polymorphisms (SNPs). SSRs are 1–6 base pair (bp) long tandem repeats, and are particularly valuable for their high level of polymorphism, reproducibility, cost effectiveness, potential for automation, and ease of transfer among laboratories [3,19,22]. They also require smaller amounts of DNA, relative to other marker systems [23]. SSR markers have been repeatedly shown to be one of the most powerful DNA-based methods for use in genetic studies in multiple crop species [24]. SNPs are the most abundant types of DNA markers in any organism [6]. They are easily amenable to automation and thus have been used for high throughput genotyping, cultivar identification, and QTL analysis [6,25–28]. Hop germplasm managers and breeders need multiple, adaptable fingerprinting methods that can distinguish siblings and resolve relationships among cultivars and between the cultivated and wild gene pools.

Hop has recently benefited from the advances in the genomic revolution. More than 1,000 SSR markers have been isolated, and they include genomic [29–32] and genic SSRs [3,5,33,34]. SSR markers in hop have been a valuable tool for genotype identification [19,29,35], parentage analysis [3,35], diversity assessments [5,8,36], linkage mapping [5,37], marker-assisted selection [7,38], and phylogenetic and evolutionary studies [18,39,40]. The SSR marker HI-AGA7 has been used to determine sex [7,41]. The HI-AGA7 marker has also been used for genotyping because it is highly polymorphic and has high allelic diversity [7,8,19,42].

High throughput genotyping techniques using SNPs have been of great benefit for hop breeding and genetics. Common techniques that have been used in hop include diversity array technology (DArT) [37,42], genotyping-by-sequencing (GBS) [6,25,43–45], and kompetitive allele specific PCR (KASP) [46]. Many SNP genotyping methods can be expensive to perform on small sample sizes, but have proven their value in linkage mapping, QTL analysis, and genome wide association studies [6,37,44,47,48]. In hop, SNP markers were used for genetic

analysis of downy mildew resistance [48,49], powdery mildew resistance [50]), sex determination [43,51], short stature [47], and drought stress tolerance [51]. Interest abounds in using SNPs for cultivar identification [6,27,28,45,46]. Yamauchi [28] used three hop cultivars to identify four SNP-containing regions from transcriptome sequences obtained from Sanger sequencing to characterize 21 hop cultivars. Matthews et al. [27] and Van Holle et al. [46] used GBS to discriminate among 178 and 56 commercial cultivars, respectively; while Jiang et al. [46] used 12 KASP markers to distinguish 16 hop cultivars. Sanger and Illumina-based sequencing for detection of these previously identified SNPs for cultivar identification are relatively expensive and the results are challenging to analyze. A cost-effective and easy to perform and evaluate SNP-based fingerprinting tool is needed in hop.

The objective of this study is to develop easy-to-use, cost effective SSR-based and SNP-based fingerprinting sets for hop that can distingish European and WNA accessions including cultivars. The development of these DNA tests will allow hop genebanks and breeding programs to establish core collections of hop germplasm and ensure the correct identity of the accessions in the collections.

## Methods

### Plant material and DNA extraction

Young actively growing hop leaf tissue was collected from 354 samples from the NCGR, 249 samples from the FSCR Breeding Program, and 26 samples from UNL (S1 Table). Tissue of the UNL accessions, was shipped overnight on ice to the NCGR in Corvallis, Oregon. Approximately 30–50 mg of tissue from each accession were sampled into a 96-well plate format. Duplicate samples were collected for SSR and KASP genotyping. Tissue for SSR genotyping was flash-frozen in liquid nitrogen. Samples were stored at -80˚C until DNA extraction. Prior to extraction, samples were ground using a mixer mill (MM 301; Retsch International, Hann, Germany). DNA was extracted using a modified DNA extraction protocol [52] for the E-Z 96 Plant DNA extraction kit (Omega BioTek, Norcross, GA, USA). DNA was quantified with a Tecan Infinite M Plex multimode plate reader (Tecan Group Ltd, Zürich, Switzerland) and diluted to 3 ng/µL. Tissue for KASP genotyping was shipped after the addition of silica to each well using the Plant Sample Collection kit (according to the manufacturer's recommendations) to LGC Biosearch Technologies, Hoddesdon, UK, where DNA extraction and genotyping was conducted.

### SSR genotyping

To develop an SSR fingerprinting DNA test, 43 primer pairs (S2 Table) were chosen. The 43 SSRs were selected based on the following criteria: 1) SSRs containing long core repeats with a base motif equal to or greater than 3 base pairs, 2) high allele diversity based on published reports, 3) ease of scoring, and 4) if they had been previously used by collaborators Claudia Wiedow at Plant and Food Research, New Zealand (Claudia.Wiedow@plantandfood.co.nz) and Josef Patzak at Hop Research Institute Co. Ltd., Czech Republic (patzak@chizatec.cz) in April, 2019. SSRs identified by Koelling et al. [33] were given the 'K' prefix. The 43 SSRs were tested individually in a panel of 16 diverse hop samples (S1 Table), consisting of 'Alliance', 'Horizon', 'Crystal', 'Scarlet', 'Dana', 'BitterGold', 'Backa', 'Talisman', 'Smooth Cone', 'Wye Challenger', 'Southern Brewer', 'Shinshuwase', 'Tardif de Bourgogne', 'Swiss Tettnanger', 'Santiam', and KAZ-067 (S1 Table), using an M13 labeling method [53]. The reactions were conducted in 15 µL volumes using the GoTaq Flexi PCR kit (Promega, Madison, WI, USA.). Reactions consisted of 3 µL of 5×GoTaq buffer, 1.2 µL of 2.5 mM dNTPs, 1.2 µL of 25 mM $MgCl_2$, 0.075 µL of GoTaq, 0.18 µL of 10 µM forward primer, 0.75 µL of 10 µM reverse primer,

0.75 μL of 10 μM fluorescently labeled M13 primer, 6.4 μL of PCR grade water, and 1.5 μL of DNA at a concentration of 3 ng/μL. Reactions were amplified in an Eppendorf Gradient thermocycler (Eppendorf Inc., Westbury, NY, USA) using a program consisting of an initial denaturation at 94˚C for 3 min; 10 cycles of denaturation at 94˚C for 40 s, annealing for 1.5 min starting at 55˚C and decreasing by 1˚C per cycle, and extension at 72˚C for 30 s; 20 cycles of denaturation at 94˚C for 40 s, annealing at 52˚C for 45 s, and extension at 72˚C for 45 s; 10 cycles of denaturation at 94˚C for 40 s, annealing at 53˚C for 45 s, and extension at 72˚C for 45 s; followed by a 30 min hold at 72˚C. Following completion of the program, the amplicons remained at 4˚C until removed from the thermocycler. Reaction success was assessed via 2% agarose gel electrophoresis. Amplicons from successful reactions were pooled, and alleles were separated via capillary electrophoresis using a GeneScan 600 LIZ dye size standard on an Applied Biosystems SeqStudio Genetic Analyzer (Thermo Fisher Scientific Inc., Waltham, MA, USA). The allele visualization and analysis were performed with GeneMarker software (version 3.0.0, SoftGenetics, LLC., State College, PA, USA) to determine allele sizes, and evaluate each SSR for absence of PCR artifacts, ease of scoring, and polymorphism. The final SSRs were selected for high polymorphism, ease of scoring (1-easy to 0.5-medium), and multiplexing ability–lack of overlapping alleles and contained the sex-linked SSR HI-AGA7 [7].

Forward primers for the SSRs selected to make up the fingerprinting set were fluorescently labeled (Thermo Fisher scientific Inc. Waltham, MA, USA) for multiplex amplification. An iterative optimization process was conducted in the testing panel to identify optimal primer concentrations and to confirm that primers did not interact. Reactions were conducted in 15 μL volumes consisting of 8.3 μL of TypeIT PCR Master Mix (Qiagen NV, Germantown, MD, USA), 1.5 μL of PCR grade water, 1.7 μL of primer mix, and 3.5 μL of DNA at a concentration of 3 ng/μL. Primer concentrations varied within the 1.7 μL of the primer mix in each iteration. Thermocycler program consisted of an initial denaturing at 95˚C for 5 min; 9 cycles of denaturation at 95˚C for 30 s, annealing for 1.5 min starting at 65˚C and decreasing by 1˚C per cycle, extension at 72˚C for 30 s; 29 cycles of denaturation at 95˚C for 30 s, annealing at 55˚C for 1.5 min, and extension at 72˚C for 30 s; followed by a 30 min hold at 60˚C. Following completion of the program, the amplicons remained at 4˚C until removed from the thermocycler. Optimized reactions using the labeled primers were performed on 32 accessions from two bi-parental populations supplied by the FSCR (S1 Table) to assess the ability of the SSR multiplex to distinguish full siblings. The SSR multiplex was then used to genotype the remaining 354 samples from the NCGR, the remaining 217 samples from the FSCR Breeding Program, and the 26 samples from UNL (S1 Table).

## KASP SNP genotyping

To develop a set of SNPs used for differentiation amongst genotypes, we performed GBS on a set of 326 *H. lupulus* var. *lupulus* genotypes, including male and female accessions, located at the USDA-ARS hop breeding program in Corvallis, Oregon. GBS library preparation used the ApeK1 cutter enzyme and sequencing was performed on an Illumina 3000 as outlined by Elshire et al. [54]. The 'Cascade' draft genome by Padgitt-Cobb et al. [55] was used as the reference genome for sequence alignment and identification of SNPs. SNP calling was performed with the TASSEL [56] 3.0 pipeline (for TASSEL scripts see Data Availability) using the default filter settings. This resulted in an initial set of SNPs that were further filtered with the TASSEL GUI [56] (version 5.2.49) to a 2X minimum depth with the "SetLowDepthGenotypesToMissing" argument and iteratively optimized to obtain a 90% genotype minimum presence and 75% taxa minimum presence with the "filter genotype table sites" and "filter genotype table taxa" arguments, respectively. SNPs were further filtered to have a minor allele frequency of

0.20 and a maximum missing percent of 10%. This set of SNPs were filtered to remove all markers that had any missing data. Using JMP Genomics software (version 9.0, JMP Genomics, SAS Institute, Cary, NC, USA), Index of probability of Hardy-Weinberg equilibrium, PIC value, heterozygosity and MAF averages were calculated for the filtered SNPs with the "Genetic Marker Statistics" and "Marker Properties" routine and sub-routine processes, respectively. Any markers that had a 0.50 average or greater for all four indices was selected for further analysis. Using a PERL-base script "MinimalMarker" by Fujii et al. [57], a final set of markers was identified that differentiated the original 326 accessions in the GBS data. The SNP and DNA sequences 200 bp upstream and 200 bp downstream of each SNP marker were submitted to LGC Biosearch Technologies for their design and validation as KASP markers. Sixty-three samples from the NCGR and 127 samples from FSCR were genotyped with the 28 KASP SNP assays (S1 and S3 Tables), including repeated controls to determine their efficacy at differentiating among genotypes as well as consistency in identifying clonal propagules as identical. Raw allele calls for the 28 assays were received from LGC, and cluster plots were viewed using the SNPviewer software (version 1.99, LGC Biosearch Technologies, Hoddesdon, UK). Each assay was evaluated, and markers were removed if they were not easily binned into three defined clusters or had greater than 20 percent missing data.

## Cluster analyses and diversity assessment

Cluster analyses were conducted using the R packages "ape" [58] (version 5.4–1) and "poppr" [59,60] (version 2.8.6). The raw allele calls for each SSR were exported from the GeneMarker software (SoftGenetics, LLC., State College, PA, USA), and converted to "poppr" format (e.g., alleles for each marker separated by ":"). The allele data were converted into a genind object with null alleles coded as "NA" with the "df2genind" function [59,60]. With the "recode_polyploid" function, unknown alleles (i.e., masking of null alleles or unknown dosage configurations) were coded as "000" [59,60]. Since *Humulus* contains diploid, triploid, and tetraploid accessions, Bruvo's distance was chosen for its ability to calculate microsatellite genetic distances irrespective of ploidy level [61]. Thus, the "bruvo.boot" command [59,60] was used to produce a neighbor joining tree with the "njs" algorithm with an infinite allele, allele-sharing model [62] by setting the "replen" for all samples to 0.001 as implemented in Metzger et al. [62] and using bootstrap support of 2,000 permutations as implemented in "ape" [58].

Individuals were classified into two groups for clustering: 1) cultivated (CULT), and 2) wild North American (WNA). Cultivated consisted of European and European Hybrid germplasm. Population richness, intra group diversity, expected heterozygosity, and evenness were calculated for the 629 accessions of CULT and WNA samples with the "poppr" function as implemented in "poppr" [59,60]. Multi-locus genotypes and expected multi-locus genotypes were used to evaluate population richness. Simpson's index was used to evaluate intra group diversity [63]. This index provides an estimation of the probability that two randomly selected genotypes are different. For each fingerprinting set and marker, the number of observed alleles and observed genotypes were counted; Nei's expected heterozygosity [64], Simpson's index [63], and evenness were calculated at the allelic and genotypic level using the "locus_table" function as implemented in "poppr" [59,60].

Null allele frequencies and confidence intervals were calculated for all markers in all diploid individuals. A custom Python script was created to transform the "poppr" formatted file into Genepop format. This entailed concatenating all marker alleles separated by ":" (e.g., 120:130 to 120130), adding "000" to single allele calls (e.g., 120 to 000120), and converting NA null alleles to "000000" (e.g., NA to 000000). The Genepop file was imported into the "nulls" function as implemented in the R package "genepop" [65] (version 1.1.7).

## Evaluating population structure

A principal coordinate analysis was conducted using Bruvo's distance measures with the "cmdscale" function as implemented in the "stats" R package [66] (version 4.0.2) to determine population structure of the 629 NCGR, FSCR, and UNL samples (S1 Table). To improve the identification of the number of true clusters, we followed recommendations of Chae and Warde [67] and Chae and Il [68] for clustering multiple binary variables in large genetic data sets. The optimal number of clusters was identified by visually inspecting the total within sum of squares plot on the principal coordinates with the fviz_nbclust function [69] (version 1.0.7) in R. To confirm that the optimal number of clusters were selected, the "majority rule" was performed by evaluating over 30 indices as implemented in the R package "Nbclust" [70] (version 3.0). Partitioning around medoids (PAM) was performed with the "pam" function on the principal coordinates with the "cluster" R package [71] (version 2.1.0). Population structure was also assessed using the sparse non-negative matrix function (sNMF) algorithm as implemented in the R package "LEA" [72,73] (version 3.0.0). The sNMF algorithm avoids the Hardy-Weinberg equilibrium assumptions and presumes that the genetic data originate from the admixture of K parental populations, where K is unknown. The "snmf" function returns estimates of ancestry proportions for each multi-locus genotype in the sample [72].

## Pedigree analysis

The matrix of allele counts from the genind object generated by the R package "poppr" was transformed following Rodzen et al. [74]. This entailed converting heterozygous and homozygous (1 and 2) alleles to presence of dominant allele (1) and null alleles (0) to absence of dominant allele (2). This transformation converted individual polyploid phenotypes at a $k$-allele codominant locus to diploid phenotypes at $k$ dominant "loci" [75]. Because limited resources exist for pedigree analysis in polyploids, the Rodzen et al. [74] conversion allows polyploid data to work with pedigree software designed for a diploid dominant marker system. The transformed data was parsed into three sub-sample types based on analysis: offspring, candidate males, and candidate females. The three sub-type files were imported into the software package "COLONY" [76] (version 2.0.6.6). COLONY configurations were set to the following parameters: polygamy for both males and females, inbreeding mating, without clones because duplicate genotypes were removed, dioecious, diploid, full likelihood estimates because it was the most accurate, and hard likelihood precision. The remaining parameters were set to default. COLONY states that the full likelihood (FL) model is the most accurate. Each parent was evaluated, and parents with pairwise likelihood (PL) under 90% were eliminated and not considered in parent PL unless a FL estimate was provided.

## Comparison of the SSR fingerprinting set to the KASP SNP assay

Performance of the SSR fingerprinting set was compared to the KASP assay in 190 samples that were genotyped in both assays. Dendrograms were produced similarly as in the cluster analysis, except neighbor joining trees were produce using the "aboot" function in the R package "poppr" [59,60] using Prevosti's distance for similar comparison [77]. Because the SNP calls are bi-allelic, "NA" values were set to be ignored with the "missing" argument within the "aboot" function. The resulting dendrograms were compared to determine how well each genotyping method distinguished all unique accessions. To confirm that differences observed between the SSR fingerprinting set and KASP assay were due to performance and not sampling error, DNA extracted by LGC Biosearch Technologies was shipped back to the NCGR overnight on dry ice. To eliminate sampling error, the samples that grouped differently in the two dendrograms; the DNA was genotyped with the SSR fingerprint set. If those re-sampled

accession resulted in the same discrepancies, the corresponding samples from LGC were genotyped with the SSR fingerprint set. If a sample was found to have sampling error, the SSR dendrogram was updated with fingerprint data from both the NCGR and LGC (respectively, N or L were added as a suffix to indicate the source of DNA). Population richness, intra group diversity, expected heterozygosity, and evenness were calculated for both the SSR fingerprinting set and the KASP assays for cultivated and WNA accessions with the "poppr" function as implemented in "poppr" [59,60]. Population richness, diversity, Nei's expected heterozygosity and evenness were also calculated for both fingerprinting sets at the allelic and genotypic level using the "locus_table" function within the R package "poppr" [59,60].

## Results

### SSR fingerprinting set development and optimization

Of the 43 SSRs evaluated in the 16-member testing panel (S1 Table), six failed to produce a PCR product, and 18 SSRs were difficult to score due to the presence of amplification artifacts or the inconsistent appearance of alleles between replications (ease of scoring: No PCR product, 0 or 0.5; S2 Table). A 9-SSR fingerprinting set was constructed from primer pairs that were polymorphic and easy to score in the testing panel (Fig 1). When optimized it distinguished the 16 individuals in the test panel, and it consisted of a 200 μL reaction volume that contained 0.40 μM HI-AGA7 6-FAM, 0.50 μM primer 1 NED, 0.40 μM primer 6 VIC, 0.50 μM K10315910 6-FAM, 3.00 μM pigtailed K10315931 VIC, 0.50 μM K10316016 VIC, 0.63 μM pigtailed K10316221 NED, 0.63 μM K10315842 6-FAM, 0.63 μM ACA 1-K9-3 PET. A pigtail, consisting of a single "GTTT", was added to 5' end of the reverse primer for K10315931 and K10316221, to reduce split peaks and ensure consistent amplicon size [78].

Subsequent use of this 9-SSR set distinguished each sibling from the two sets of 16 individuals in the bi-parental populations HQ2015023 and HQ2015034 (Fig 2). Each offspring had one allele in common with the female parent but no alleles in common with the genotyped male parent, except for two samples, indicating either an error in sampling the male parent or a mistake in pollination (S4 Table). To eliminate sampling error, the putative male parent was resampled, and genotyping was repeated but generated the same results.

### KASP assay development and validation

Genotyping-by-sequencing with the TASSEL 3.0 pipeline in the 326 *H. lupulus* var *lupulus* genotypes initially identified approximately 1.1 million SNPs (S5 Table). Filtering using the

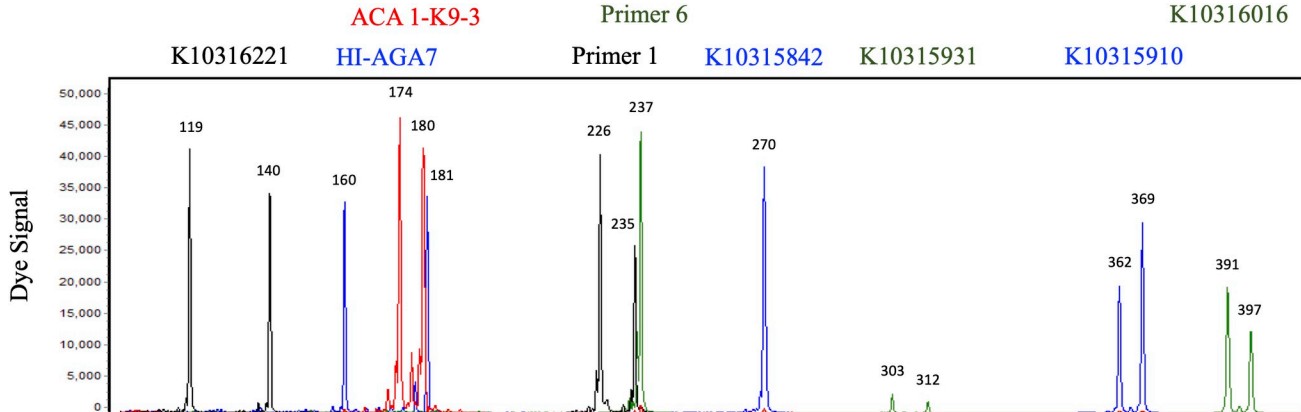

**Fig 1. Electropherogram example for the 9-SSR fingerprinting set in 'Cascade'.** For each SSR, the size of the peaks is provided in bp and the color indicates the dye label (NED in black, PET in red, FAM in blue and VIC in green).

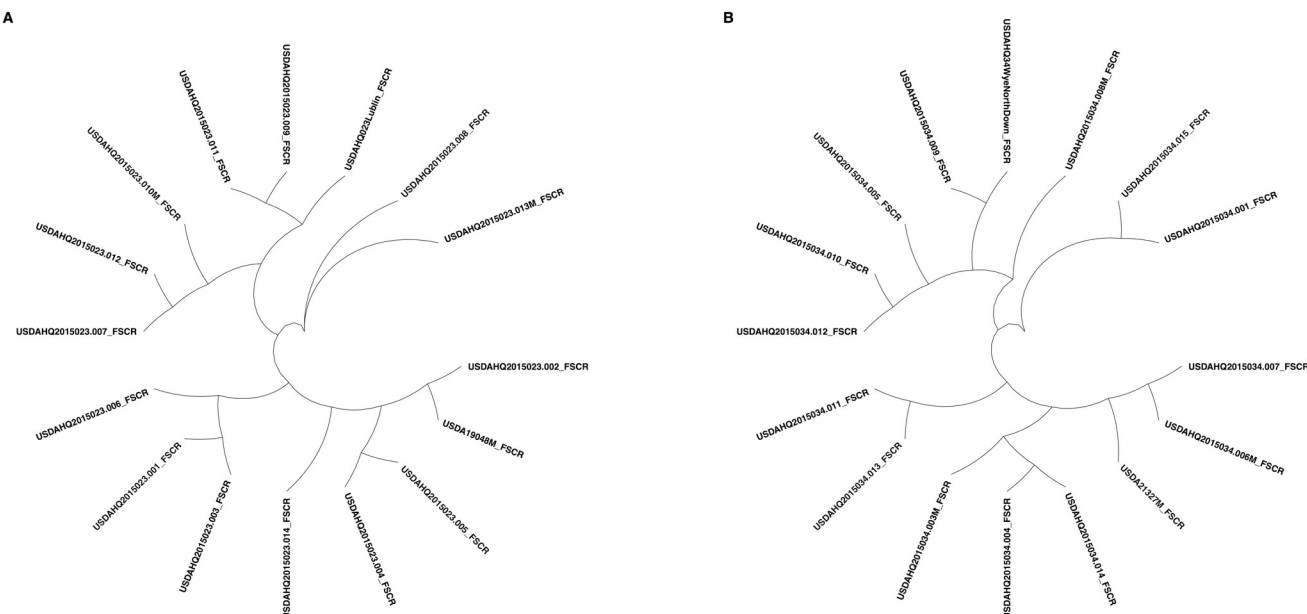

**Fig 2.** Radial dendrogram of bi-parental populations HQ2015023 (A) and HQ2015034 (B) evaluated with the 9-SSR fingerprinting set.

TASSEL GUI identified SNPs in the following order: 1) 35,633 SNPs that had 2X minimum depth, 90% genotype minimum presence, and 75% taxa minimum presence, 2) 6,676 SNPs with a minor allele frequency of 0.20 and maximum missing percentage of 10%, 3) 117 SNPs with no missing data in all 326 accessions. The JMP Genomics software identified 717 SNPs that had a greater than 0.5 average for Index of probability of Hardy-Weinberg equilibrium, PIC value, Heterozygosity, and MAF. Using the "MinimalMarker" script, a final set of 28 SNPs were selected that differentiate the original 326 genotypes.

Out of the 28 KASP markers evaluated, only 25 markers had three well-defined clusters and were included in the analyses, while markers 006996F_176695 and 278F_370690 were mono-morphic and marker 008065F_133678 only had two homozygous and well-defined clusters. These three markers were removed prior to data analysis (S3 Table). Hueller Bitterer CHUM 746 was the only sample that had over 20 percent missing data, possibly indicating low quality DNA. Genotype data were obtained for all samples in 18 assays, and of the remaining five assays 0.5–2.6% of the samples failed to amplify. The KASP markers 63_359415F and 5449F_211727 failed to amplify in all the WNA germplasm.

To validate the 28 KASP markers, the X and Y bi-allelic calls were compared in 63 samples that were in common between the 28 KASP assays and the original GBS data (S6 Table). Mis-matches between GBS and KASP were classified into three main groups: 1) GBS heterozygous allele calls that were homozygous with KASP (e.g., X:Y to X:X and X:Y to Y:Y), 2) GBS homozy-gous calls that were homozygous for the opposite allele with KASP (e.g., X:X to Y:Y and Y:Y to X:X), and 3) GBS homozygous calls that were heterozygous with KASP (e.g., X:X to X:Y and Y:Y to X:Y) (S7 and S8 Tables). The majority of mismatch calls between GBS and KASP were hetero-zygous to homozygous (41.95%) and homozygous to heterozygous call types (45.85%, S8 Table). GBS homozygous calls that were homozygous for the opposite allele with KASP were only found in 12 individuals (S7 and S8 Tables) and accounted for 12.20 percent of all mismatches. These 12 individuals each had greater than 20 percent incongruity within the 28 loci and accounted for the majority of total genotype mismatches. When excluding these 12 samples, there was a 96.29 percent agreement between the 28 original GBS and KASP allele calls (S7 Table).

## Evaluation of the 9-SSR fingerprinting set in diverse germplasm

Because the 9-SSR multiplex distinguished each sibling in the two bi-parental populations reported earlier, it was used to genotype 354 accessions from the NCGR collection, 217 additional individuals from the FSCR Breeding Program, and 26 individuals from UNL (S1 Table). Of the 629 samples that were genotyped, 105 accessions were known to originate from North America, the other 524 samples were assumed to be cultivated with European pedigrees or a hybridization between European and WNA pedigrees.

Eighty-nine sets of samples had identical fingerprints and were assembled into genotype groups (S9 Table). These groups included: samples representing the same cultivar under different designations between the NCGR and the FSCR; some of the same cultivars donated or obtained by the NCGR from different sources and given different local numbers (indicated by the prefix CHUM for Corvallis Humulus); clonal selections; and heat-treated clones (indicated by htcl) (S9 Table). Each group of synonyms was scrutinized to determine the likelihood of identical genotypes. Clonal selections were confirmed or are proposed as indicated by the same genotypes, and included clones of:

**'Ahil':** 'Francia B8', received in 2008 from Serbia as a clone of 'Ahil';

**'Backa':** 'Backa E', a clonal selection of 'Backa' originating from the Backa regions in former Yugoslavia;

'Elsasser' and 'Francia B5', clonal selections made in France and possible selections of a 'Backa' type clone [79];

**'Brewer's Gold':** 'Gold Brauer' which may be a German selection of 'Brewer's Gold' as its name would suggest;

**'Bullion'**: 'Bullion 10A', a heat-treated meristem culture of Bullion;

**'Fuggle':** 'Fuggle 57', 'Fuggle-H', 'Fuggle-N', Tetraploid 'Fuggle', 'Savinja Golding', 'Styrian Golding', and 'Yugoslavia Golding' as selections developed from the original 'Fuggle' (USDA 19209) [17,79]; 'Star', an older Belgian variety, as an early 'Fuggle' clone;

**'Golding'**: 'Canterbury Golding', 'Bramling', 'Cobbs Golding', 'East well Golding', and 'East Kent Golding', known as the Goldings and previously established as clones [17];

**'Hersbrucker'**: The Hersbruckers as clonal selections from the original 'Hersbrucker' (USDA 21673) [79]; and

K-51, an unknown 'Hersbrucker' selection.

**'Northern Brewer'**: 'Blue Northern Brewer', a clone of 'Northern Brewer' that has blue toned stems and leaves;

**'Saazer'**: Osvald Saazer Clone 114, 31, 72, and 72Y, clonal selections of 'Saazer' (USDA 21077) made by Osvald at Zatec in 1924 [17]; Saazer 36vf and Saazer 38vf, heat treated meristems of 'Saazer' developed by Dr. Skotland at Washington State University; 'Aromat', 'Blato', 'Lubelska', 'Lublin', 'Lucan', 'Nadwislanka', 'Sirem', 'Spalter', 'Zultan', and 'Universal' reported as Saazer types or alternatively marketed as Saazer [3,35,79–81];

'Precoce de Bourgogne', a clonal selection of 'Saazer' made in France [80]; and Htcl 1/22, Htcl 1/23, Htcl 4/12, Htcl 4/6, Htcl 6/4, and Htcl 7/18, as heat-treated clones from the Saazer group.

**'Shinshuwase'**: 'Kirin II', 'Kirin No. 5 Y'1, and 'Golden Star' are clones of 'Shinshuwase' that were developed by the Kirin Brewery Company and the Sapporo Brewery in Japan, respectively;

**'Talisman'**: 'Pocket Talisman', a clone of 'Talisman' with shorter stature and reduced leaf size;

**'Tettnanger':** 'Swiss Tettnanger', 'Tettnanger European', and 'Tettnanger US', from different sources and clonal selections of the original 'Tettnanger' (USDA 61021 and USDA 21015),

developed in the Tettnang region of Germany as a clonal selection of an old landrace [79]. Previous sources have identified the original 'Tettnanger' as a 'Saazer' clone [3,79,80,82];

**USDA 21612**: Htcl 2/23, a heat-treated clone of USDA 21612

**'Wuerttemberger':** 'Wuerttemberger 48' and 'Wuerttemberger 49', clones from the original 'Wuerttemberger' (USDA 21682), an older German landrace that could be an early clone of 'Backa' that is related to a Serbian style hop [79];

In each synonym group, we then identified samples that are not known to be clones of other accessions in that synonym set and re-sampled them to exclude sampling error and identify incorrectly labeled accessions. Accessions that were confirmed as incorrectly labeled included:

**'Alliance' CHUM 167.005, and 'Alliance' CHUM 167.001** had the same genotypes as 'Scarlet' CHUM 1597.001, and USDA 9418.44 representing 'Scarlet' at the FSCR.

**'Apolon' CHUM 1482** had a unique fingerprint that didn't match 'Apolon' CHUM 162 and USDA 21051 representing 'Apolon' at the FSCR, suggesting that 'Apolon' CHUM 1482 is not true-to-type.

**'Blisk' CHUM 1485.001, 'Canterbury Whitebine' CHUM 1556.001, 'Groene Bel' CHUM 1480.001, 'Hersbrucker' CHUM 1498.001, 'Hersbrucker-G' CHUM 199.002, 'Oregon Cluster' CHUM 1533.001, and 'Tardif de Bourgogne' CHUM 192.003** had identical fingerprints to each other and to that of clones in the 'Backa' group. 'Blisk' CHUM 1485.001 should have the same fingerprint as that of 'Blisk' CHUM 166.002. 'Canterbury Whitebine' gave rise to the first Goldings and should match the other 'Golding' clones. 'Hersbrucker' and 'Hersbrucker-G' should have identical fingerprints to that of the 'Hersbrucker' clones. 'Oregon Cluster' is a WNA hop and 'Tardif de Bourgogne' CHUM 192.003 should have the same genotype as that of USDA 21169 FSCR and 'Herbrucker 8' as previously reported by Bassil et al. [3]. We believe that all seven of these accessions to be mis-labeled.

**'Celeia' CHUM 1490.001, 'Hueller Hallertauer Mittelfrueh' CHUM 1507.001, and USDA 21084** had the same genotypes as that of the 'Saazer' clones, indicating they are mis-labeled. 'Celeia' CHUM 1490.001 and 'Hueller Hallertauer Mittelfrueh' CHUM 1507.001 plants are in neighboring pots and adjacent to several 'Saazer' clones, possibly indicating a plant contamination due to a mixture of genotypes in the same pot. USDA 21084 is supposed to be a wild selection from Yugoslavia.

**'Early Promise' CHUM 1518.001, 'Early bird Gold' CHUM 1517.001, Osvald Saazer Clone 1267 CHUM 1543.001, 'Wuerttemberger' CHUM 920.004, and 'Zatecki Cerveni' CHUM 1539.001** had the same genotypes as that of the 'Fuggle' clones. According to Salmon, 'Early promise' is a mass selection from Wye College resulting from a cross between an English hop (D9) with an English hop (B1) [83]. 'Early bird' (aka 'Early bird Golding') has already been established as a Golding clone of the original 'Golding' selected in 1790 [17]. Osvald Saazer Clone 1267 is most likely Osvald Clone 126, a known Fuggle-type selected by Dr. Karel Osvald during his work from 1927 to 1948 [84]. 'Wuerttemberger', should be the original 'Wuerttemberger', a Serbian style hop accession. 'Zatecki Cerveni' is a 'Saazer' type hop, whose name is translated as "Saazer Red". The plants representing all but Osvald Saazer Clone 1267 are in pots next to two plants of 'Fuggle' (CHUM 1519.001 and CHUM 1520.001), possibly indicating plant contamination.

**'Groene Bel' CHUM 1553.001** had the same genotype as four accessions of 'Brewer's Gold' from different sources (CHUM 1512.001, CHUM 30.002, CHUM 1493.001, and from the FSCR).

***Humulus lupulus* var. *neomexicanus* Colorado 2–1 (WildAmerican) CHUM 8.003** had the same genotype as two plants of 'Santiam' from different sources (USDA-ARS NCGR and

the USDA-ARS FSCR). 'Santiam' was developed by Haunold at the FSCR in the 1990's and is most likely true-to-type.

***Humulus lupulus* var. *neomexicanus* Colorado 3–1 (WildAmerican) CHUM 106.003 and 'Hueller Bitterer' CHUM 126.003** had identical fingerprints to that of 'Early Cluster E2'. Fingerprints of the plant CHUM 126.003 representing 'Hueller Bitterer' was different from that of two plants from different sources, CHUM 1506.001 and CHUM 746.002, representing the same cultivar. This would indicate that *Humulus lupulus* var. *neomexicanus* Colorado 3–1 (WildAmerican) and 'Hueller Bitterer' CHUM 126.003 are mis-labeled clones of 'Early Cluster E2'; parentage analysis is needed to determine if 'Early Cluster E2' is the true representative of this group.

**K-14 CHUM 1564.001, 'Fuggle 125' CHUM 1587.001, K-15 CHUM 1565.001, K-65 CHUM 1578.001, and K-95 CHUM 1585.001** had identical fingerprints and are housed next to each other. 'Fuggle 125' is expected to have the same fingerprint as the original 'Fuggle', as it is a 'Fuggle' clone. This potentially would suggest that all five samples are errors.

**K-62A CHUM 1576.001, K-23 (Robusta) CHUM 1567.001, K-6 (Aroma) CHUM 1562.001**, had the same genotype as '**Ringwood Special' CHUM 1561.001** and are housed right next to each other at the USDA-ARS NCGR. The "K" samples and 'Ringwood Special' were donated in 2008 from Serbia and little is known about them. This would suggest that all four samples could potentially be mis-labeled in the USDA-ARS NCGR collection and further work is needed to confirm identity.

**K-90 CHUM 1583.001, and Kirin No. 5A1 CHUM 1557.001** had the same genotype as two accessions of 'Hueller Bitterer' from different sources in the NCGR collection (CHUM 746.002 and CHUM 1506.001). 'Kirin No. 5A1' is a clone of 'Shinshuwase' developed by Kirin Brewery Company. The plants representing K-90, Kirin No. 5A1, and 'Hueller Bitterer 'CHUM 1506.001 are in neighboring pots in the screenhouse, possibly indicating contamination.

**K-98 CHUM 1586.001 and K-78 CHUM 1582.001** have identical fingerprints and are housed next to each other.

**Late Cluster Seedling** had the same genotype as two plants of 'Calicross' from different sources (CHUM 175.002 and CHUM 1534.001);

**'Pacific Gem' CHUM 803.002 and 'Alpharoma' CHUM 627.002** had identical fingerprints. This was unexpected because 'Alpharoma' was developed in the late 1970's by Dr. Frost, and 'Pacific Gem' was released in 1988 by Dr. Fost's successor Dr. Beatson at the Plant and Food Research in Motueka, New Zealand. Since 'Pacific Gem' and 'Alpharoma' have the same parents ('Smooth Cone' x Unknown male (OP)), parentage analysis will not resolve the correct identity of this genotype. The NCGR will have to obtain these two samples from another source to determine identity.

**'Saazer' CHUM 1544.001, 'Cerera' CHUM 1491.001** had the same genotypes as the three accessions of 'Celeia' (CHUM 889.005, CHUM 889.004, and CHUM 1491.001) at the NCGR. Since this 'Saazer' plant (CHUM 1544.001) is the only plant with a genotype that is different from those in the Saazer group, we believe this plant to be mis-labeled and it will be discarded. The genotype of 'Cerera' CHUM 1491.001 is different from that of the USDA 21612 representing this accession at the FSCR.

**'Scarlet' CHUM 1597.004, and 'Scarlet' CHUM 1597.007** had the same genotypes as that of the two accessions of 'Alliance' CHUM 167.008, and USDA 66050, from the NCGR and the FSCR, respectively. 'Scarlet' was developed by Dr. Haunold at the FSCR and 'Alliance' was developed at Wye College. Since the genotype of the CHUM 1597.004 and CHUM 1597.007 representing 'Scarlet' are different from that of CHUM 1597.001 and that of the USDA 9418–

44 plant also representing this cultivar, we believe that CHUM 1597.004 and CHUM 16597.007 are mis-labeled and should be discarded.

'Sunshine' CHUM 140.002 had the same genotype as 'Eastern Green Kirin' C827 (CHUM 929.002 and USDA 21700 from the FSCR).

'Pride of Kent' CHUM 148.002 had an identical genotype to that of 'Sunshine' represented by the USDA 21281 accession at the FSCR. Parentage analysis is needed to determine if these two plants represent 'Sunshine' or 'Pride of Kent'.

When evaluating UNL samples, all had unique fingerprints, except for five samples. 'Joplin 3' and 'Saaz offtype' had the same fingerprint as 'Cascade', indicating that these two samples were 'Cascade' clones. 'Joplin 4' had an identical fingerprint to 'Chinook', suggesting that 'Japlin 4' is a clone of 'Chinook'. 'MHPsd11' and 'MHPsd10' had identical genotypes (S9 Table). The remaining samples were uniquely identified with the 9 SSR fingerprinting set.

## Diversity parameters of the 9-SSR fingerprinting set

Population richness showed that there were more multi-locus genotypes in the cultivated samples (331) than in the known wild WNA samples (94) (Table 1). When accounting for sample size, the expected number of multi-locus genotypes were the same between the WNA and cultivated samples (94). Intra group diversity was slightly higher for the cultivated samples (0.997) than the WNA samples (0.989). These slight differences could be due to differences in sample size. Expected heterozygosity and evenness was roughly the same for the cultivated samples (0.685 and 1) and the WNA samples (0.686 and 1).

For all samples, the mean number of alleles was 11.78 with HI-AGA7 having the highest number of observed alleles (30), and K10316016 having the lowest number of observed alleles (3) (Table 2). The average number of genotypes was 40.67 with HI-AGA7 having the highest number of genotypes (109) and K10316016 having the lowest number of genotypes (5). Mean Simpson's diversity and Nei's expected heterozygosity were identical, 0.73 and 0.85 for alleles and genotypes, respectively. Simpson's allelic and genotypic diversity were highest at HI-AGA7 (0.88 and 0.96, respectively) and lowest at K10315910 (0.48 and 0.60, respectively).

**Table 1. Summary statistics for all unique cultivated (CULT) and wild North American (WNA) hop accessions with the 9-SSR fingerprinting set and in unique genotypes of the 190 accessions evaluated with the 9-SSR and 25 KASP sets.**

| Population | Sample Number | Multi-locus genotypes | Expected multi-locus genotypes | Simpson's Index | Nei's expected heterozygosity | Evenness |
|---|---|---|---|---|---|---|
| All samples 9 SSR | | | | | | |
| CULT | 331 | 331 | 94 | 0.997 | 0.685 | 1 |
| WNA | 94 | 94 | 94 | 0.989 | 0.686 | 1 |
| Total | 425 | 425 | 94 | 0.998 | 0.729 | 1 |
| 190 sample comparison– 9 SSR | | | | | | |
| CULT | 135 | 135 | 16 | 0.993 | 0.673 | 1 |
| WNA | 16 | 15 | 15 | 0.93 | 0.615 | 0.967 |
| Total | 151 | 150 | 16 | 0.993 | 0.7 | 0.996 |
| 190 sample comparison—25 KASP assays | | | | | | |
| CULT | 136 | 135 | 10 | 0.993 | 0.45 | 0.996 |
| WNA | 4 | 4 | 4 | 0.75 | 0.326 | 1 |
| Total | 140 | 139 | 10 | 0.993 | 0.451 | 0.996 |

Includes number of unique genotypes (sample number), number of expected multi-locus genotypes, number of multi-locus genotypes, intra group diversity represented by Simpson's Index, Nei's expected heterozygosity, and population evenness. Simpson's index, Nei's 1978 gene diversity (expected heterozygosity), evenness, and null allele frequency range from 0 to 1.

**Table 2. Locus and genotype summary statistics for all samples evaluated with the 9-SSR set and in the 190 accessions genotyped with both marker systems, 9-SSR set and 25 KASP assays.**

| Locus | Observed allele size range | Observed alleles (no.) | Simpson's index (allelic) | Nei's 1978 gene diversity (allelic) | Evenness (allelic) | Observed genotypes (no.) | Simpson's Index (genotypic) | Nei's 1978 gene diversity (genotypic) | Evenness (genotypic) | Null Allele Freq.[a] | Null Allele CI[a] |
|---|---|---|---|---|---|---|---|---|---|---|---|
| All samples 9 SSR | | | | | | | | | | | |
| HI-AGA7 | 157–246 | 30 | 0.88 | 0.88 | 0.63 | 109.00 | 0.96 | 0.97 | 0.51 | 0.00 | - |
| K10315842 | 261–285 | 8 | 0.77 | 0.77 | 0.84 | 29.00 | 0.90 | 0.91 | 0.72 | 0.02 | 0.0000 0.0587 |
| K10315910 | 356–369 | 5 | 0.48 | 0.48 | 0.74 | 7.00 | 0.60 | 0.60 | 0.67 | 0.26 | 0.2117 0.3101 |
| K10316221 | 116–174 | 14 | 0.82 | 0.82 | 0.76 | 51.00 | 0.94 | 0.95 | 0.69 | 0.00 | - |
| Primer 1 | 226–260 | 11 | 0.79 | 0.79 | 0.76 | 44.00 | 0.91 | 0.92 | 0.61 | 0.00 | - |
| Primer 6 | 225–249 | 7 | 0.75 | 0.75 | 0.90 | 21.00 | 0.88 | 0.89 | 0.81 | 0.12 | 0.0790 0.1717 |
| K10315931 | 297–321 | 9 | 0.72 | 0.72 | 0.78 | 29.00 | 0.86 | 0.86 | 0.65 | 0.00 | - |
| K10316016 | 391–397 | 3 | 0.50 | 0.51 | 0.97 | 5.00 | 0.65 | 0.66 | 0.93 | 0.08 | 0.0267 0.1389 |
| ACA 1-K9-3 | 156–265 | 19 | 0.83 | 0.83 | 0.66 | 71.00 | 0.94 | 0.94 | 0.55 | 0.00 | - |
| Mean | - | 11.78 | 0.73 | 0.73 | 0.78 | 40.67 | 0.85 | 0.85 | 0.68 | - | - |
| 190 sample comparison—9 SSRs | | | | | | | | | | | |
| HI-AGA7 | - | 21 | 0.84 | 0.84 | 0.64 | 47.00 | 0.93 | 0.94 | 0.57 | - | - |
| K10315842 | - | 6 | 0.71 | 0.72 | 0.85 | 16.00 | 0.86 | 0.87 | 0.78 | - | - |
| K10315910 | - | 3 | 0.46 | 0.46 | 0.73 | 4.00 | 0.58 | 0.58 | 0.72 | - | - |
| K10316221 | - | 11 | 0.80 | 0.80 | 0.72 | 32.00 | 0.93 | 0.94 | 0.72 | - | - |
| Primer 1 | - | 11 | 0.79 | 0.79 | 0.77 | 32.00 | 0.91 | 0.91 | 0.64 | - | - |
| Primer 6 | - | 6 | 0.70 | 0.71 | 0.86 | 14.00 | 0.85 | 0.86 | 0.82 | - | - |
| K10315931 | - | 5 | 0.68 | 0.68 | 0.79 | 12.00 | 0.83 | 0.84 | 0.78 | - | - |
| K10316016 | - | 2 | 0.50 | 0.50 | 0.99 | 3.00 | 0.59 | 0.59 | 0.85 | - | - |
| ACA 1-K9-3 | - | 14 | 0.79 | 0.80 | 0.64 | 39.00 | 0.92 | 0.93 | 0.61 | - | - |
| Mean | - | 8.78 | 0.70 | 0.70 | 0.78 | 22.11 | 0.82 | 0.83 | 0.72 | - | - |
| 190 sample comparison—25 KASP assays | | | | | | | | | | | |
| 000089F_317093 | - | 2 | 0.44 | 0.44 | 0.89 | 3.00 | 0.60 | 0.60 | 0.90 | - | - |
| 000265F_1374141 | - | 2 | 0.50 | 0.50 | 1.00 | 3.00 | 0.63 | 0.64 | 0.93 | - | - |
| 000571F_797591 | - | 2 | 0.47 | 0.47 | 0.94 | 3.00 | 0.61 | 0.62 | 0.91 | - | - |
| 001644F_470045 | - | 2 | 0.50 | 0.50 | 1.00 | 3.00 | 0.66 | 0.66 | 0.98 | - | - |
| 002381F_371663 | - | 2 | 0.40 | 0.40 | 0.83 | 3.00 | 0.56 | 0.57 | 0.86 | - | - |
| 003591F_48935 | - | 2 | 0.44 | 0.44 | 0.89 | 3.00 | 0.59 | 0.59 | 0.89 | - | - |
| 008739F_51678 | - | 2 | 0.47 | 0.47 | 0.94 | 3.00 | 0.63 | 0.64 | 0.93 | - | - |
| 000115F_1068460_Y | - | 2 | 0.50 | 0.50 | 1.00 | 3.00 | 0.66 | 0.66 | 0.98 | - | - |
| 001589F_351349_R | - | 2 | 0.34 | 0.34 | 0.75 | 3.00 | 0.51 | 0.51 | 0.83 | - | - |
| 002878F_402249_W | - | 2 | 0.44 | 0.44 | 0.89 | 3.00 | 0.58 | 0.59 | 0.89 | - | - |
| 003765F_59789_M | - | 2 | 0.50 | 0.50 | 1.00 | 3.00 | 0.61 | 0.61 | 0.88 | - | - |
| 23F_2218589_C_A | - | 2 | 0.48 | 0.48 | 0.95 | 3.00 | 0.60 | 0.60 | 0.89 | - | - |
| 63_359415F_C_T | - | 2 | 0.49 | 0.49 | 0.97 | 3.00 | 0.65 | 0.65 | 0.97 | - | - |
| 122F_1097361_T_A | - | 2 | 0.49 | 0.49 | 0.99 | 3.00 | 0.62 | 0.63 | 0.91 | - | - |
| 198F_1478737_C_T | - | 2 | 0.47 | 0.47 | 0.94 | 3.00 | 0.60 | 0.60 | 0.89 | - | - |
| 559F_410159_G_T | - | 2 | 0.29 | 0.29 | 0.68 | 3.00 | 0.44 | 0.44 | 0.71 | - | - |
| 1037F_456722_A_G | - | 2 | 0.50 | 0.50 | 0.99 | 3.00 | 0.62 | 0.63 | 0.91 | - | - |

(*Continued*)

**Table 2.** (Continued)

| Locus | Observed allele size range | Observed alleles (no.) | Simpson's index (allelic) | Nei's 1978 gene diversity (allelic) | Evenness (allelic) | Observed genotypes (no.) | Simpson's Index (genotypic) | Nei's 1978 gene diversity (genotypic) | Evenness (genotypic) | Null Allele Freq.[a] | Null Allele CI[a] |
|---|---|---|---|---|---|---|---|---|---|---|---|
| 1345_507801_T_C | - | 2 | 0.43 | 0.43 | 0.88 | 3.00 | 0.58 | 0.58 | 0.89 | - | - |
| 1371F_177062_C_A | - | 2 | 0.50 | 0.50 | 0.99 | 3.00 | 0.57 | 0.57 | 0.83 | - | - |
| 1808F_307171_A_T | - | 2 | 0.48 | 0.48 | 0.96 | 3.00 | 0.59 | 0.59 | 0.87 | - | - |
| 2181F_220335_A_G | - | 2 | 0.41 | 0.41 | 0.85 | 3.00 | 0.58 | 0.58 | 0.87 | - | - |
| 2814F_80926_T_G | - | 2 | 0.50 | 0.50 | 1.00 | 3.00 | 0.60 | 0.61 | 0.87 | - | - |
| 2925F_116637_C_T | - | 2 | 0.50 | 0.50 | 1.00 | 3.00 | 0.60 | 0.61 | 0.87 | - | - |
| 5441F_117067_T_G | - | 2 | 0.47 | 0.47 | 0.94 | 3.00 | 0.63 | 0.63 | 0.93 | - | - |
| 5449F_211727_A_C | - | 2 | 0.27 | 0.27 | 0.67 | 3.00 | 0.42 | 0.42 | 0.70 | - | - |
| Mean | - | 2 | 0.45 | 0.45 | 0.92 | 3.00 | 0.59 | 0.59 | 0.88 | - | - |

Includes locus name, observed number of alleles, observed number of genotypes, and allelic and genotypic estimates for Simpson's index, Nei's 1978 gene diversity (expected heterozygosity), and evenness, null allele frequency, and null allele confidence interval. Allele size range is provided for SSRs in the 629 samples. Null allele frequencies and confidence intervals were only evaluated in diploid samples. (-) indicates not evaluated or not applicable.

[a] Null allele frequencies and confidence intervals calculated for diploid individuals.

Nei's expected allelic and genotypic heterozygosity were also highest at HI-AGA7 (0.88 and 0.97, respectively) and lowest at K10315910 (0.48 and 0.60, respectively). Average allelic and genotypic evenness were 0.78 and 0.68, respectively. K10316016 had the highest (0.97 and 0.93) and HI-AGA7 had the lowest (0.63 and 0.51) allelic and genotypic evenness, respectively.

Genepop identified null alleles in four of the nine SSR markers. Marker K10315910 had the highest null allele frequency of 0.26 and K10315842 had the lowest null allele frequency of 0.02 (Table 2). Primer 6 and K10316016 had intermediate frequencies of 0.12 and 0.08, respectively.

## Population structure and clustering based on the 9-SSR fingerprinting set

Hierarchical clustering using Bruvo's distance identified distinct groups that separated primarily according to geography and pedigree (Fig 3). Hierarchical clustering grouped samples into three main clades (S1 Fig). Three clusters were also observed with K-medoids clustering (Fig 3), except for some slight variations. Cluster 1 was considered primarily as continental European hops containing 'Backa', 'Saazer', 'Northern Brewer', 'Golding', German selections, and related germplasm. Cluster 2 represents mostly USA developed hops with WNA parentage along with actual WNA samples, except for *Humulus lupulus* var. *lupulus* KAZ098. Cluster 3 represents hop lines derived primarily from English hop lines containing 'Brewer's Gold', 'Fuggle', 'Hersbrucker', some North American germplasm, Wild Yugoslavian 3/3 germplasm and related germplasm. A discriminant analysis of the data had the three clusters separating primarily into WNA germplasm and cultivated germplasm (Fig 4). Most of the variance was explained by the first two principal coordinates (47.3%). Additionally, three groups were also observed using the sNMF algorithm (Fig 5), supporting the number of groups identified by both hierarchical clustering and partitioning around medoids.

## Parentage analysis

COLONY was used to confirm parentage (S10 Table) and sibship (S11 Table) in the two tested FSCR bi-parental populations, HQ2015023 and HQ2015034. USDA HQ023Lublin was

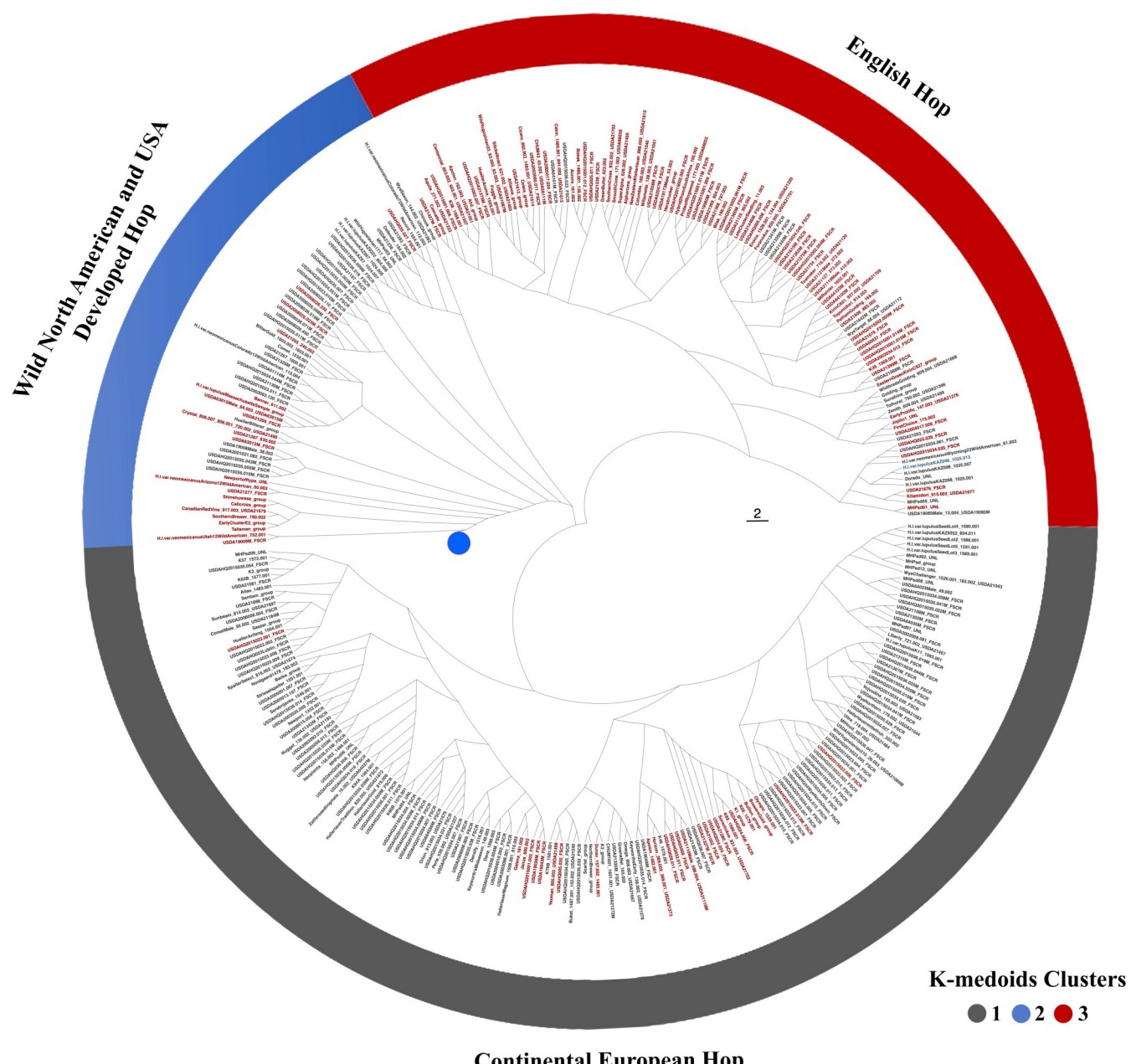

**Fig 3. Radial dendrogram of 629 accessions evaluated with the 9-SSR fingerprinting set.** K-medoids identified three clusters and clustering results for each accession are displayed in different colors for each group. Wild North American samples are collapsed and represented by the blue dot in the dendrogram (see S1 Fig for full dendrogram).

confirmed as the female parent for population HQ2015023 with an FL probability of 0.7165 and a 95% PL confidence level. No candidate males were identified with the FL model. The PL model identified six males with only one male having a confidence level over 90%. This male was USDA 19048M and was associated with the offspring USDA HQ2015023.002. This supports our previous findings that USDA HQ2015023.002 is the only possible offspring of USDA HQ023Lublin and USDA 19048M (S4 Table). Within the HQ2015023 population, the maximum likelihood (ML) model identified 5 possible full sibling relationships (S11 Table). For

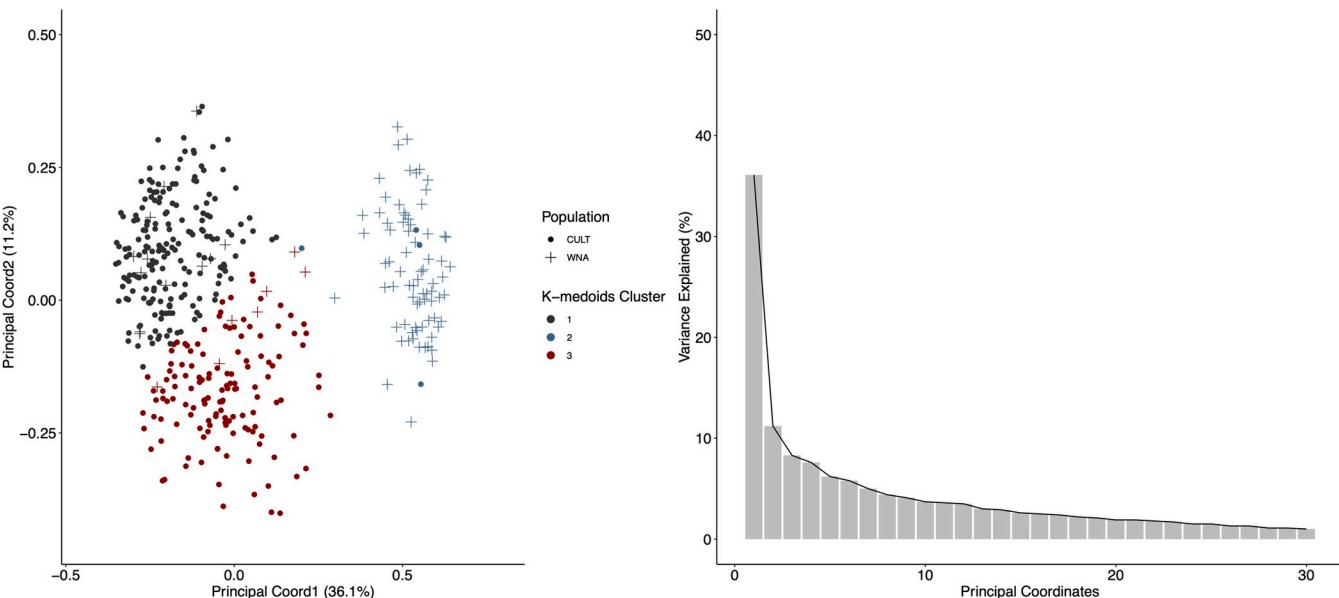

**Fig 4. Principal coordinate analysis of 629 accessions using the 9-SSR fingerprinting set.** Population structure using k-medoids is reflected on the first two principal coordinates. Explained variance of the two principal coordinates is shown as a percentage in the axis titles. k-medoids clusters and population for each group are displayed as different symbols and colors, respectively. A plot displaying the variance explained by each principal coordinate is presented.

population HQ2015034, USDA HQ34WyeNorthdown was confirmed as the female parent with an FL probability of 0.5948 and a 95% PL confidence level. The FL model didn't identify any candidate males. The PL model identified five males, but all had a confidence level lower than 75%. The ML model identified five possible full sibling relationships (S11 Table).

Nine sets of samples had non-matching fingerprints between the NCGR and the FSCR (S12 Table). These nine sets were re-collected and genotyped with the nine SSRs to eliminate

**Humulus Data (K = 3)**

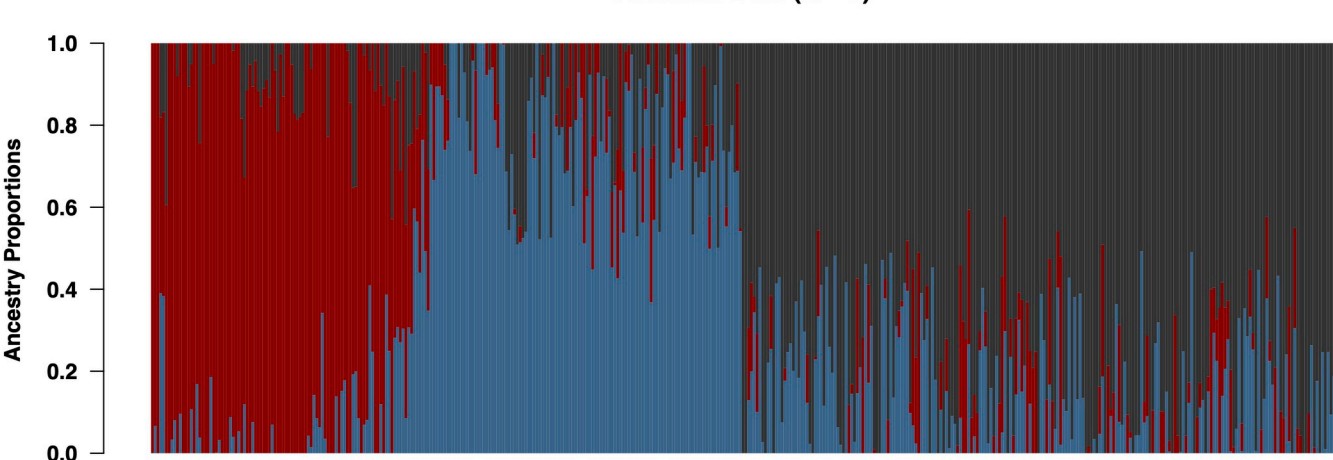

**Fig 5. Structure analysis of 629 accessions using the 9-SSR fingerprinting set using the sNMF algorithm.** Three subpopulations were identified, and ancestry proportions are displayed as different colors.

sample collection error or contamination during extraction, but the results generated were the same. Parentage (S10 Table) and sibship (S11 Table) analyses were performed using COLONY on the nine sets to determine which individual in each set was likely the correct individual based on reported pedigree. Cultivars evaluated in COLONY are the following:

1. **Atlas** = 'Brewer's Gold' x Yugoslavian wild male 3/3; a tetraploid cultivar generated by colchicine (S1 Table). Brewer's Gold was only designated as the female parent for USDA 21052 representing 'Atlas' at the FSCR with a FL probability of 0.9854 and a PL confidence level of 95%. COLONY revealed PL males for both 'Atlas' CHUM 1483.001 from the NCGR and USDA 21052, but confidence levels were under 90%. No male parent was suggested for FL. The Yugoslavian wild male 3/3 (USDA 21087M) genotype available at the FSCR was not a candidate male. Parent PL was high for both CHUM 1483.001 (0.9521) and USDA 21052 (0.9799), and sibship showed that both 'Atlas' samples belong to two separate families. This indicates that USDA 21052 is probably the true-to-type 'Atlas'.

2. **'Aurora'** = 'Northern Brewer' x Yugoslavian wild male TG. Yugoslavian wild male TG was not available. Only Aurora CHUM 161.002 had Northern Brewer as the female parent with 0.9309 FL probability and a 95% PL confidence level. No FL or PL male candidate was suggested for either Aurora CHUM 161.002 from the NCGR or its synonym USDA 21053 from the FSCR. Parent PL probability was highest at 0.9203 for USDA 21053 having two unknown parents. Sibling relationship revealed that Aurora CHUM 161.002 and USDA 21053 belonged to two separate families with a probability of 0.9999, indicating that Aurora CHUM 161.002 is most likely true-to-type.

3. **'Banner'** = 'Brewer's Gold' x Unknown male (OP). 'Brewer's Gold' was confirmed as the female parent only for Banner CHUM 611.002 from the NCGR with FL probability of 0.9869 and a PL confidence level of 95%. Unexpectedly, COLONY indicated that 'Banner' CHUM 611.002 and USDA 21287 representing 'Banner' at the FSCR FL male parent was USDA 21139M with 0.9933 probability. PL for the male parent was under 90% for both samples. Parent PL probability was 0.849 and 0.8503 for USDA 21287 and Banner CHUM 611, respectively. Sibship indicated that both 'Banner' samples are from two different families, suggesting that 'Banner' CHUM 611.002 is most likely true-to-type.

4. **'Blisk'** = 'Atlas' x Yugoslavian male 1/9. We used 'Atlas' USDA 21052 from the FSCR as the female parent in COLONY. In both 'Blisk' samples, CHUM 166.002 from the NCGR, and USDA 21238 from the FSCR, 'Atlas' was suggested as a female parent, but with low probability FL probability (0.2583). Only 'Blisk' CHUM 166.002 had a PL confidence level of 95%. COLONY identified a PL candidate male for 'Blisk' CHUM 166.002 with lower than 90% confidence level. No FL or PL male parent was suggested for USDA 21238. Parent PL was highest for 'Blisk' CHUM 166.002 and sibship indicated that these two 'Blisk' genotypes are from two different families, suggesting that 'Blisk' CHUM 166.002 is likely true-to-type.

5. **'Bobek'** = 'Northern Brewer' x Wild Yugoslavian male TG. In CHUM 1486.001 and CHUM 139.002, 'Northern Brewer' was indicated as a female parent only for PL with a confidence level of 95%. COLONY didn't designate a FL or PL male parent for 'Bobek' CHUM 1486.001 and CHUM 139.002. A PL candidate male was suggested for USDA 21239 representing FSCR's 'Bobek', but the confidence level was less than 90%. Sibship suggested that the two 'Bobek' samples are highly related and could be full siblings. Parent PL probability was highest (0.8069) for USDA 21239 having two unknown parents, suggesting that 'Bobek' CHUM 1486.001 and CHUM 139.002 are true-to-type.

6. The cultivar **Bullion** doesn't have parents represented in our data set. To determine which 'Bullion' is true-to-type, we performed a sibship analysis in COLONY with sibling 'Brewer's Gold'. 'Bullion' and 'Brewer's Gold' resulted from a cross between Wild Manitoba BB1 and an unknown male (OP). COLONY sibship analysis indicated that all the 'Brewer's Gold' clones in the NCGR collection are full siblings with the 'Bullion' clones in the collection. USDA 21056 representing 'Bullion' from FSCR was shown to belong to a separate family group, suggesting that this 'Bullion' clone is not true-to-type.

7. **'Dunav'** = 'Northern Brewer' x 'Yugoslavian male Sx-502'. COLONY indicated that the 'Dunav' accessions at the NCGR from different sources (CHUM 157.002 & CHUM 1465.001) and 'Dunav' represented by USDA 21081 at the FSCR have equal FL probability (0.4084) and PL confidence (95%) that 'Northern Brewer' is the female parent. Yugoslavian male Sx-502 was not sampled. Thus, COLONY didn't indicate a candidate male for FL or PL. Parent PL probability was also equally likely (0.9912) for both samples. Sibship between the 'Dunav' samples showed that both samples are highly related and potentially full siblings. The NCGR will have to request a sample from another source to determine which 'Dunav' sample is the correct individual.

8. **'Southern Brewer'** = 'Fuggle N' x unknown male (OP) backcross. No female candidates were designated for FL and PL for either sample; thus, 'Fuggle N' was not a candidate parent for either sample. COLONY indicated male parents for both 'Southern Brewer' CHUM 190.002 at the NCGR and USDA 21187 representing 'Southern Brewer' at the FSCR, but they had PL confidence levels lower than 90%. For CHUM 190.002 and USDA 21187, Parent PL was the highest with unknown male and female parents at 0.9795 and 0.5072, respectively. Sibship analysis revealed that both 'Southern Brewer' samples came from different families. The NCGR will have to obtain a sample from another source to identify the true-to-type 'Southern Brewer'.

9. **'Toyomidori'** = 'Northern Brewer' x USDA 64103M. For 'Toyomidori' CHUM 914.003 from the NCGR, COLONY indicated that 'Norther Brewer' is the female parent with FL probability of 0.6991 and a PL confidence level of 90%; and USDA 64103M is the male parent with FL probability of 0.578. Parent PL probability was 0.9563. USDA 21676, representing FSCR's 'Toyomidori', didn't have a candidate male or female for FL or PL. Parent PL probability for both unknown parents was 0.6599. The sibling relationship between the two 'Toyomidori' samples showed that they belonged to two different families. This suggests that 'Toyomidori' CHUM 914.003 matches its reported pedigree and is true-to-type.

To confirm that 'Early Cluster E2', 'Pride of Kent', and 'Sunshine' are true-to-type, parentage (S10 Table) and sibship (S11 Table) analysis were performed in COLONY:

'Early Cluster E2' is most likely a somatic mutation of an 'Oregon Cluster' or 'Californian Cluster' [80]. The names Oregon and Californian Cluster represent a number of different Cluster cultivars. Since parentage is unknown in 'Early cluster E2' we ran parentage and the sibship analysis in COLONY with all available samples and all available males and females, respectively. COLONY indicated that the best sibship configuration was between 'Early Cluster E2' and 'Talisman'. For Parentage, COLONY indicated that 'Calicross', 'Talisman', 'Green Bullet', 'Smoothcone', 'Southern Cross', 'Shinshuwase', and 'Joplin 1' were candidate female parents of 'Early Cluster E2', although highly unlikely due to the years the cultivars were created. The cultivar Calicross was developed by a 'California Cluster'/Fuggle Seedling cross (S1 Table). 'Talisman' was produced from a 'Late Cluster'/unknown male (OP). 'Smooth Cone' was developed by a 'California Cluster'/unknown male (OP) cross. 'Green Bullet' and 'Southern Cross' are offspring of 'Smooth Cone'. 'Shinshuwase' has 'White Vine' as a grandparent and 'Joplin 1' is a WNA hop.

The highest FL probabilities were with 'Calicross' (0.5454) and 'Talisman' (0.3372). 'Talisman' was developed by Dr. Romanko at the Agriculture Experiment Station in Moscow, Idaho in 1959, and 'Early Cluster E2' was developed by Dr. Skotland at the Irrigated Agriculture Research and Extension Center in Prosser, Washington in the late 1950's [80]. 'Calicross' was developed in 1960 by Dr. Rudi Roborgh at the New Zealand Horticultural Research Centre [17]. Since 'Early Cluster E2' was most likely developed within a year or two of 'Calicross' and 'Talisman', they are probably not the parents. This does suggest that they are all highly related and have close 'Cluster' parentage, and 'Early Cluster E2' could be true-to-type.

The genotype of 'Pride of Kent' represented by USDA 21280 at the FSCR was different from that of 'Pride of Kent' CHUM 148.002 at the NCGR. 'Pride of Kent' CHUM 148.002 had the same fingerprint as USDA 21281 representing 'Sunshine' at the FSCR. This left uncertainty as to which accession represented 'Pride of Kent' and which one was 'Sunshine'. 'Pride of Kent' resulted from a cross between 'Brewer's Gold' and an unknown male (OP) while 'Sunshine' was derived from Female OS99 x unknown male (OP). In COLONY, we tested which offspring was a descendent of 'Brewer's Gold' as it was the only parent available in our data set. COLONY indicated that 'Brewer's Gold' was only a candidate female to USDA 21280 with FL probability of 0.9997 and a PL confidence level of 95%. No males were indicated for either sample for FL and PL. Parent PL probability was high at 0.9334 for USDA 21280 and 0.9513 for Pride of Kent CHUM 148.002. Sibship showed that both samples belonged to two different families. This suggests that USDA 21280 is 'Pride of Kent' and Pride of Kent CHUM 148.002 was mis-labeled and is 'Sunshine'.

To confirm COLONY results, raw allele calls from each sample were visually compared to alleles of the female parent, when available (S13 Table). A sample was considered a possible offspring of the female parent if it had at least one allele in common with any of the female's alleles at a locus. The analysis per female are as following:

For 'Northern Brewer' as the parent, USDA 21081, 'Dunav' (CHUM 157.002, 'Dunav' CHUM 1465.001), 'Toyomidori' CHUM 914.003, 'Aurora' CHUM 161.002, and 'Bobek' CHUM 1486.001 all had at least one allele in common with the female parent at all nine SSR loci. Confirming the COLONY results, USDA 21676 ('Toyomidori') and USDA 21239 ('Bobek') did not share alleles at one locus with their reported parent, while USDA 21053 ('Aurora') did not share alleles at two loci with its reported female parent (S13 Table).

'Banner' CHUM 611.002 from the NCGR, USDA 21052 ('Atlas'), and USDA 21280 ('Sunshine') had at least on allele in common with 'Brewer's Gold' at all SSR loci, confirming that the alternative genotypes represented by USDA 21287 ('Banner'), 'Atlas' CHUM 1483.001, and individuals from the 'Sunshine' group were not derived from 'Brewer's Gold' and are not true-to-type.

Using USDA 21052 as the 'Atlas' female parent, showed that USDA 21238 ('Blisk') had at least one allele in common with 'Atlas'. 'Blisk' CHUM 166.002 had alleles at one locus (Primer 6) that were not shared with the reported female parent, which did not agree with the results from COLONY. 'Blisk' CHUM 166.002 had up to three alleles at a given locus, consistent with its triploid chromosome number and origin from a tetraploid by a diploid parent. 'Blisk' accessions from different sources are needed to determine which individual 'Blisk' accession is true-to-type.

'Southern Brewer' CHUM 190 and USDA 21187 had at least two loci that were not in common with the female parent 'Fuggle', supporting results from COLONY that both samples could not have been derived from 'Fuggle N'. The NCGR will also need to obtain 'Southern Brewer' from a reliable source and conduct parentage analysis to identify the accession that represents the true-to-type cultivar.

## Comparison of the 9-SSR fingerprint set to the KASP assay

A total of 190 hop samples were genotyped with both the 9-SSR fingerprinting set and the 25 KASP assays. Hierarchal clustering using Prevosti's distance produced 15 sets of synonyms with the 9-SSR fingerprint set and 17 sets of samples that had identical fingerprints with the KASP assays (Fig 6 and S14 Table). To determine the cause of the discrepancies, we re-sampled the 16 accessions that were synonyms with only one of the DNA tests. These included 'Aromat' CHUM 1538.001, *H. lupulus* var. *neomexicanus* Colorado 2–1 CHUM 8.003, and the 3 'Burlington' N2 samples and the 11 wild accessions in the 'WildNorthAmerican' synonym group identified by the KASP assays.

Those re-sampled accessions and the corresponding DNA sent back from LGC Biosearch Technologies were genotyped with the 9-SSR fingerprint set. The resulting 9-SSR genotypes were compared to each other and to that independently collected for SSR analysis of the 629 hop samples. The genotype of *H. lupulus* var. *neomexicanus* Colorado 2–1 CHUM 8.003 DNA from LGC was identical to that of 'Santiam' while that of *H. lupulus* var. *neomexicanus* Colorado 3–1 CHUM 106.003 matched that of *H. lupulus* var. *neomexicanus* Wyoming 3–1 CHUM 91.002 (Fig 6). *H. lupulus* var. *neomexicanus* OCJ-58 CHUM 1385.001 and 'Aromat' CHUM 1538.001 were identified as unique samples. The 9-SSR genotypes of the remaining 12 samples were identical indicating that the correct plant was sampled, and the two additional synonym sets (Burlington N2, and Wild North American) were not distinguished with the KASP assay.

Group names were given to the synonym sets to reduce genotype redundancy (S14 Table) and only unique genotypes were used to calculate genetic diversity estimates in cultivated and WNA germplasm in both fingerprinting sets (Table 1). A larger number of multi-locus genotypes was generated by the 9-SSR fingerprinting set (150) than that observed in the KASP fingerprinting set (139) (Table 1). WNA samples had more multi-locus SSR genotypes (15) than SNP genotypes (4) using the KASP assay, while cultivated samples had identical numbers of genotypes (135). When accounting for population size, the WNA samples still had the same number of multi-locus SSR and SNP genotypes, while the cultivated samples generated by the 9-SSR fingerprinting set had more genotypes (16) than the KASP fingerprinting set (10).

Simpson's index was high in cultivated samples with both marker systems, but it was higher in WNA samples with the 9-SSR set (0.930) than with the KASP markers (0.750). Total Nei's expected heterozygosity was higher in the SSR fingerprinting set (0.700) than in the KASP fingerprinting set (0.451) with the cultivated being higher (0.673 and 0.450) than WNA samples (0.615 and 0.326) in the SSR and KASP fingerprinting sets, respectively. Total evenness was high in both fingerprinting sets (0.996).

For the 9-SSR fingerprinting set the average number of alleles was 8.78 with HI-AGA7 having the highest number of alleles (21) and K10316016 with the lowest number of alleles (2) (Table 2). The average allelic Simpson's index and Nei's expected heterozygosity were identical (0.7) with HI-AGA7 having the highest (0.84) and K10315910 having the lowest (0.46) numbers for both parameters. Average allelic evenness was 0.78 and ranged between 0.64 at ACA 1-K9-3 and HI-AGA7 to 0.99 at K10316016. The average observed number of genotypes was 22.11 and ranged from 3 at K10316016 to 47 in HI-AGA7. Average genotypic Simpson's index and Nei's expected heterozygosity were similar (0.82 and 0.83, respectively) and ranged from 0.58 in K10315910 to great then 0.90 in HI-AGA7. Average genotypic evenness was 0.72 with highest evenness at K10316016 (0.85) and lowest evenness at HI-AGA7 (0.57).

For the 25 KASP assays, the average number of alleles was two as expected for bi-allelic SNPs. Average allelic Simpson's index and Nei's expected heterozygosity were the same (0.45) and the average allelic evenness was (0.92) (Table 2). Simpson's index, Nei's expected heterozygosity, and evenness were the highest in 000265F_1374141, 001644F_470045,

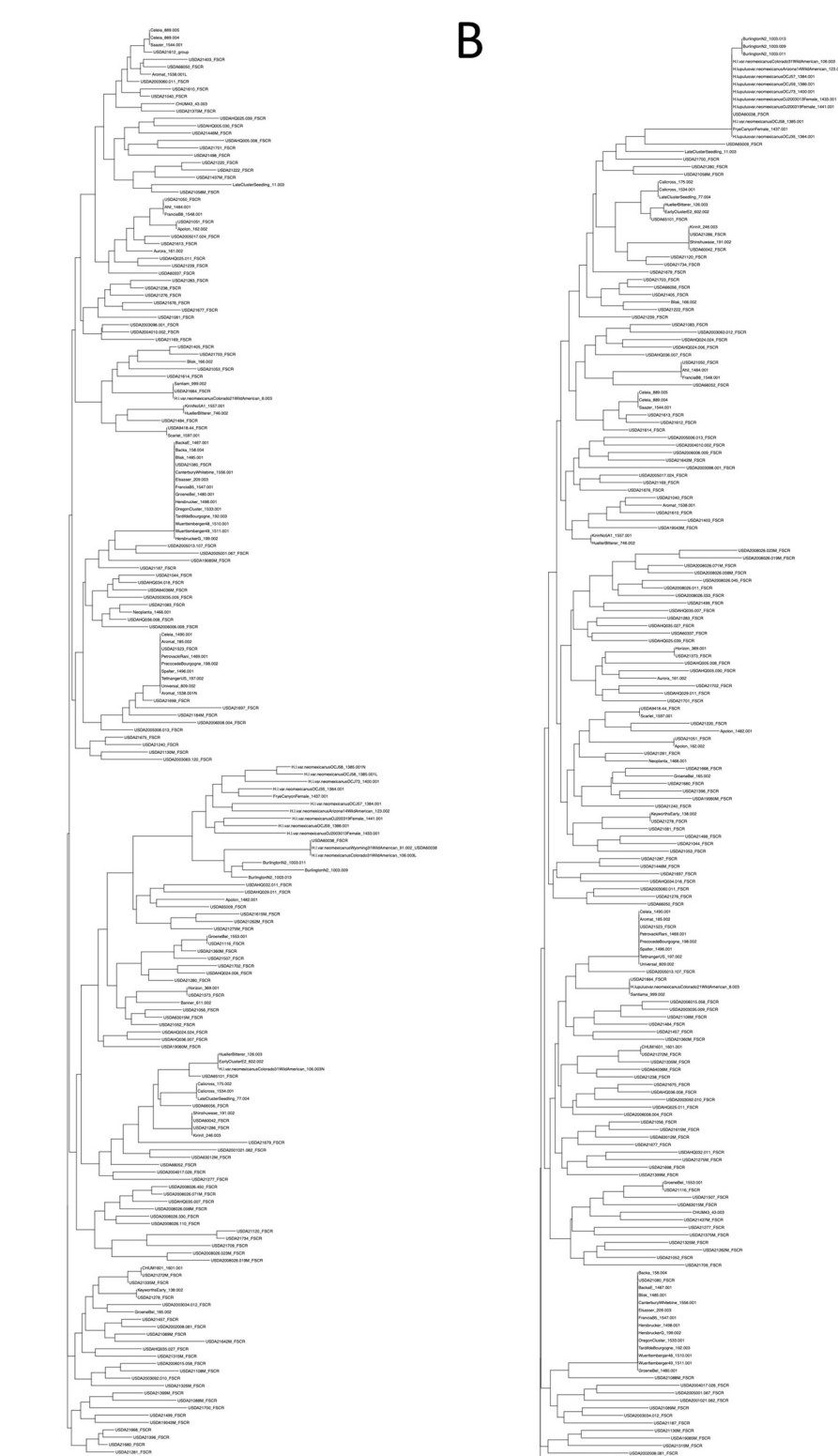

**Fig 6.** Two dendrograms based on Prevosti's distance measures in the 190 hop accessions genotyped with both marker systems, 9-SSR set (A) and 25 KASP assays (B).

000115F_1068460_Y, 003765F_59789_M, 1037F_456722_A_G, 2814F_80926_T_G, and 2925F_116637_C_T (0.50, 0.50, 1) and lowest in 559F_410159_G_T (0.29,0.29, 0.68), and 5449F_211727_A_C (0.27, 0.27, 0.67), respectively. The average number of genotypes observed were three consisting of the two homozygous and the one heterozygous genotypes. Average genotypic Simpson's index and Nei's expected heterozygosity were the same (0.59). Average genotypic evenness was 0.88. The highest genotypic Simpson's index was in 001644F_470045 and 000115F_1068460_Y (0.66) and the lowest was in 559F_410159_G_T (0.44). Genotypic Nei's expected heterozygosity and evenness were highest in 001644F_470045 and 000115F_1068460_Y (0.66 and 0.98, respectively) and lowest in 559F_410159_G_T (0.44 and 0.71, respectively).

## Sex determination

The SSR HI-AGA7 was previously found to be linked to the male sex [7,42] and has been useful in genotyping because of its high allele diversity [7,8,19,43]. HI-AGA7 was included in the 9-SSR genotyping set and used to genotype 629 individuals with known sex from the NCGR, FSCR, and UNL programs (S1 Table). We compared allele calls at this locus in accessions in common between our study and that of three previous studies that used HI-AGA7: Jakse et al. [7], Korbecka-Glinka et al. [19], and to Rodolfi et al. [43] (S15 Table). In total, 38 samples were in common with the three studies (S15 Table), including 30 samples with Jakse et al. [7], 4 with Korbecka-Glinka et al. [19], and 21 with Rodolfi et al. [43]. Allele comparison across studies in common accessions allowed us to identify the corresponding alleles that were given slightly different scores in the different studies (S16 Table).

All samples in common with Jakse et al. [7] had similar fingerprints, except for two samples: *H. lupulus* var. *neomexicanus* OCJ-25 CHUM 1355.001 and 'Hueller Aroma' CHUM 1505.001. In our study, the genotype of *H. lupulus* var. *neomexicanus* OCJ-25 CHUM 1355.001 was 221 and the same name accession was scored as 117 and 217 at HI-AGA7 (Jakse et al. [7], the later corresponding to our 221 allele). However, 117 is outside of the marker range and was most likely a PCR artifact in the Jakse et al. [7] study. The 'Hueller Aroma' samples represented at the NCGR and in Jakse et al. [7] appear to be different samples as their genotypes were different [181 & 204 (= 206 in our study), respectively]. All four samples in common with Korbecka-Glinka et al. [19] had similar fingerprints and are most likely the same individuals. Only 12 of the 21 samples in common with the Rodolfi et al. [43] had similar fingerprints. The nine cultivars with non-corresponding fingerprints at HI-AGA7 included Aurora, Cascade, El Dorado, Fuggle, Galena, Northern Brewer, Spalter Select, Wye Challenger, and Wye Target.

The HI-AGA7 primer pairs generated 30 alleles (S17 Table). Five hundred and thirteen samples were females and 115 samples were males, after confirming that the plant representing *H. lupulus* var. *neomexicanus* Colorado 2–1 CHUM 8.003 is 'Santiam' (S1 Table). Two alleles, 163 and 230, were male-specific. Allele 163 was found in cultivated accessions as in the three siblings from the HQ2015034 population and male parent USDA 21327M from the FSCR (S1 Table). Unfortunately, very few males in the HQ2015034 population inherited the 163 allele. Allele 230 was only found in a WNA accession, *H. lupulus* var. *neomexicanus* OCJ 74 CHUM 1401.001. Nine alleles were specific to females and included: 157, 169, 187, 193, 200, 233, 236, 243, and 246. The two smallest sex-specific alleles (157 in females, 163 in males) were found in the cultivated gene pool. Two of the female-specific alleles that were intermediate in size (169 and 187) were present in cultivated and WNA accessions while all larger alleles were specific to the WNA gene pool. These female-specific alleles were found in one to six accessions each.

From the shared samples with Jakse et al. [7], we were able to confirm sex-linkage of the 227 allele corresponding to 230 in our study where it was specific to the WNA accession, *H. lupulus* var. *neomexicanus* OCJ 74 CHUM 1401.001.

## Discussion

### Developing an SSR fingerprinting set

Genotyping large collections such as those at germplasm repositories and breeding programs is an essential practice for optimal management of these valuable genetic resources. However, a fast, accurate, and easy-to-use tool is needed that can distinguish the diverse cultivated and wild accessions that are conserved in these collection. Following best practices used in human forensics, we multiplexed long core repeat SSRs for accuracy and repeatability. Hop has benefited from the sequencing revolution, enabling access to thousands of SSR sequences, including long-core repeat motifs exceeding two bases [5,7,19,30–32,42,80,82]. Initially, dinucleotide- containing SSRs were used in plants because of their relative abundance and polymorphism when compared to SSRs with longer repeat motifs [84]. However, they were found to suffer from stuttering, split peaks, and binning errors [24,85,86]. Such artifacts caused difficulties in automation of allele calling, and genetic profile discrepancies among technicians and laboratories [87].

Technological advances have also progressed in PCR methods such as polymerases and multiplex mixes [88,89]. These advances have facilitated SSR multiplexing, which allows primer pairs to be simultaneously amplified in a single reaction, minimizing genotyping errors and resulting in cheaper and faster assays. For the last decade, multiplexing has proven to be a reliable technique in genotyping large collections such as in blueberry (*Vaccinium* L. species) [90], hazelnut (*Corylus avellana* L.) [91,92], blackberry (*Rubus* sp.) [93], and pear (*Pyrus* sp.) [86] at the NCGR, white Guinea yam (*Dioscorea rotundata* Poir.) [94] at the International Institute of Tropical Agriculture (IITA), and grape (*Vitis*) at the national repositories of CRA-VIT [95]. This technique continues to be an easy and cost-effective method of choice for fingerprinting large collections [90–95].

The 9-SSR fingerprinting set was first shown to be sufficient at distinguishing all siblings from two bi-parental populations (HQ2015023 and HQ2015034). This was important to demonstrate as it indicated that it could allow us to uniquely identify closely related hop cultivars.

Previously, both SNPs generated by GBS and SSRs have been shown to distinguish siblings. In hop, GBS has distinguished siblings from bi-parental populations [6,25,27]. SSRs have been reliably shown to distinguish siblings in multiple crop species [96–98]. In strawberry (*Fragaria* x *ananassa*), a 10 multiplexed SSR set distinguished 60 individuals, including siblings derived from the same parental lines [96]; in olive trees (*Olea europaea* L. subsp. *europaea*), a 12 SSR set uniquely identified 66 cultivars, including 10 sets of siblings [97]; and in pitaya (*Hylocereus* spp.) a 16 SSR set distinguished 30 individuals, including three sibling sets [98]. When compared to SSR and KASP genotyping, GBS can be more costly, bioinformatically intense, and it results in different loci between different GBS runs.

### Evaluation of the 9-SSR fingerprinting set

The 9-SSR hop fingerprinting set was also sufficient at distinguishing unique genotypes among the 629 samples, except for 89 synonym sets that were determined to be clones, sampling errors, pot contaminations, or source errors. These findings were not unexpected given hop's long cultivated history, easy vegetative propagation methods, and lack of accurate early cultivar identification techniques. Evidence of hop cultivation has been documented since the ninth century with the earliest records describing vegetative multiplication methods by division of rhizomes or the cutting of underground stems [17].

In the early years of hop domestication, superior plants were selected from the local vegetation or the unintentional establishment of a superior seedling and clonally propagated; each

selected based on adaptability to local soils and climatic conditions. This practice gave rise to the multiplication and distribution of early cultivars such as Canterbury Whitebine, Fuggle, Saazer, and Tettnang and their associated clones. Initial plant and cultivar identification methods were primarily based on morphological characteristics but were highly unreliable because morphology can change based on environmental influences such as growing conditions [8,19,20,21], and this commonly led to errors in identification.

Later identification methods sought to incorporate biochemical markers to identify individuals but were also insufficient because biochemical profiles of hop batches can fluctuate based on environmental influences, processing, and storage conditions [19,46]. Since hop has been traded for centuries without accurate identification methods, this has led to collections containing cultivars that are not true-to-type. These issues are compounded by hop's ability to be easily propagated by air layering and by stem and rhizome cuttings, making pot and yard contamination a continuous problem [6].

The most reliable methods of cultivar identification is through molecular markers as they are insensitive to growing conditions or environmental factors [19]. Still, molecular markers such as SSRs and SNPs in hop have not been able to distinguish clones [3,18,19,27,28,45,82,99]. In Henning et al. [6], in silico analysis of 32,206 high quality SNPs generated through GBS sequencing were reported to identify 7 SNP loci that can distinguish 116 hop accessions including clonal selections and sister hop lines.

As these loci were identified in silico and not converted into markers that were used to validate their distinguishing power, we believe that further study is needed to demonstrate their ability to distinguish clones is not caused by sequencing error at these loci. Clonal variations commonly arise through small or single point mutations [100,101]. Therefore, unless the markers are developed in these causal loci, clonal variants cannot be uniquely identified with common fingerprinting sets of a small number of markers. In a few studies, SSRs were reported to distinguish some variants like in sassafras (*Sassafras albidum*) [102], glabrous sarcandra herb (*Sarcandra glabra*) [103], and pear (*Pyrus communis*) [104].

Overall, the observed allele size ranges, and number of alleles in the 9-SSR testing panel of 16 diverse accessions (S2 Table) were slightly smaller than that in the full set of 629 accession (Table 2), as expected due to the increased number of samples. Out of all SSR and KASP markers, HI-AGA7 showed the highest allele diversity which is consistent with that obtained by Jakse et al. [7]. HI-AGA7 was developed by Stajner et al. [32] and subsequently used in genotyping in three additional studies [7,19,42]. When compared to Stajner et al. [32], HI-AGA7 produced a narrower allele size range (159–210), and a smaller number of alleles (13) than the 157–246 allele range and the 30 alleles in this study (Table 2).

Narrower allele size ranges (174–205, 163–193) and smaller number of alleles (5, 9) were also reported in the nine hop cultivars genotyped by Korbecka-Glinka et al. [19] and the 123 cultivated Italian, European, and American samples by Rodolfi et al. [42], respectively. Jakse et al. [7] had a comparable allele size range (157–249) and a larger number of alleles (35) in the 174 samples genotyped when compared to those obtained in our study, most likely due to the inclusion of WNA germplasm in these two studies. Primer pairs K10315842, K10315910, K10316221, K10315931, K10316016 were developed and used to genotype eight samples in Koelling et al. [33]. Primer 1 was developed by Patzak and Henychova [80] and used to genotype 135 world hop cultivars, producing 7 unique alleles. Primer 6 was developed by Patzak and Matousek [105] to characterize 11 hop samples that included WNA germplasm, and where the observed allele size range was 230–255 and the number of alleles were 5. In our study, as expected due to the larger number and diversity of the hop samples used, Primers 1 and 6 generated a slightly larger number of alleles (11 and 7, respectively) than in the original studies. Jakse et al. [7] used ACA1-K9-3 to genotype 34 hop cultivars, which resulted in a

narrower size range (191–248 as opposed to 156–265) and comparable number of alleles (20 vs. 19) to those obtained in the 629 samples in this study (Table 2).

## Validation of the 28 KASP assays

There was a high congruity between the original 25 GBS bi-allelic SNP calls and the KASP cluster calls, except in 12 individuals (S6 Table). Each of the 12 individuals had over 20% incongruity and were the only samples for which GBS was homozygous for one allele and the other allele was homozygous in the KASP assay. After excluding these 12 samples, the mismatch rate between GBS and KASP decreased from 13.04% to 3.70% (S7 and S8 Tables). Previous studies comparing KASP cluster calls to their original GBS allele calls have reported a high congruity rate with a small number of mismatches as we observed in this study [106–111]. Incongruity of the KASP and GBS genotypes in the 12 individuals is most likely due to sampling errors while the small number of genotype mismatches in the remaining samples could have resulted from sequencing error or genotype calling errors in the GBS calling pipeline.

## Diversity and population structure based on the 9-SSR fingerprinting set

Most distance estimates do not account for varying ploidy levels or dosage [61]. To overcome these limitations, Bruvo et al. [61] developed a distance calculation that took into account microsatellite mutations presuming a slipped-strand mispairing model and genome additions and losses. Bruvo's distance genome addition and loss component assumes that genomes are gained or lost over thousands of years and does not take into account that genomes such as those in hop can change from one generation to the next.

A solution to this limitation was proposed by Metzger et al. [62] by setting the repeat length parameter of Bruvo's distance to a value smaller than one nucleotide; as was tested in blackberry by Zurn et al. [93]. Like blackberry, hop can have varying ploidy levels resulting from tetraploid by diploid crosses. By changing the repeat length, the model becomes an allele-sharing model similar to Prevosti's absolute genetic distance while accounting for varying levels in mixed-ploidy populations.

Another challenging problem is clustering large binary data sets, as in genind objects, and identifying the true number of clusters. To address this problem, Chae and Il [68] and Chae and Warde [67] proposed that the cluster analysis is to be performed on the principal coordinates (PCs) and not the dissimilarity matrix itself. Chae and Il [68] concluded that the recovery levels were increased using the PCs and had a significant effect on the identification of the true number of clusters. Thus, the PCs were used to improve the identification of the true number of clusters.

Hierarchical and K-medoids clustering identified three clusters, as did sNMF based on individual ancestry coefficients. Clustering separated primarily based on cultivated versus WNA material and implies an evolutionary divergence between species based on geographic centers of origin. This is similar to what Peredo et al. [18] reported using seven SSRs to genotype 182 hop accession that included cultivars, wild European, and WNA samples. Peredo et al. [18] performed an AMOVA on two main groups ('Wild European and Cultivated' vs 'Wild American' and 'Wild European' vs 'Wild American and Cultivated') with STRUCTURE and a PCoA analysis. Based on the AMOVA analysis the 'Wild European and Cultivated' versus the 'Wild American' comparison was highly significant (12.5% of the total variance), while no statistical significance was found between the 'Wild European' versus the 'Wild American and Cultivated' comparison.

The distinction of cultivated from WNA samples is similar to what has been found in 11 previous studies performed with SSR and SNP markers

[3,5,6,26,34,36,39,40,80,81,111,112,113]. A small proportion of the WNA samples, including *Humulus lupulus* var. *lupulus* KAZ098 CHUM 1025.013, grouped with the 'Fuggle', 'Hersbrucker', Wild Yugoslavian Male 3/3, and 'California Cluster'- related material. This suggests that *Humulus lupulus* var. *lupulus* KAZ098 is either closely related to or is WNA germplasm. Differences in clustering could mostly be explained by differences according to identity-by-state (IBS) or in identity-by-descent (IBD). Hierarchical clustering tended to group samples by IBS, whereas K-medoids grouped samples based on IBD. This is most noticeable in the clustering of 'Brewer's Gold', 'Golding', and closely related samples.

When examining hierarchal clustering, 'Brewer's Gold' and closely related cultivars grouped with 'Northern Brewer' and its relatives. 'Northern Brewer' and 'Brewer's Gold' have distinctly different parentages, but share WNA parentage, suggesting clustering was grouping by IBS. K-medoids clustered 'Brewer's Gold' and closely related material with individuals that had common parentage, indicating samples were grouped by IBD. The same observations were made with the 'Goldings' and closely related samples. Hierarchical clustering grouped the 'Golding' samples and close relatives with samples that had WNA parentage as expected based on IBS. Whereas K-medoids clustered the 'Golding' samples with other samples that share a common parent and are related through IBD.

## Investigation of parentage and sibship

The mixed ploidy populations in hop pose significant problems when determining parentage with existing methods. Most tools available for parentage analysis are developed for diploids or auto-tetraploids [76,114,115]. To address this challenge, Rodzen et al. [74] proposed converting heterozygous and homozygous calls at a *k*-allele codominant locus to a presence/absence of the dominant allele. This conversion allows for a codominant marker system to become a pseudodiploid dominant marker system, regardless of ploidy level; thus, allowing the evaluation of parentage in software that is compatible with a dominant marker type system. Wang and Scribner [75] evaluated the Rodzen et al. [74] method in COLONY with the FL model and determined that regardless of inheritance mode (disomic or nondisomic) the proposed data transformation worked well and produced reliable results for both sibship and parentage assignments.

Using the 9-SSR set, we were able to determine parentage in the two bi-parental populations (HQ2015023 and HQ2015034) and the synonym sets with COLONY and confirmed relationships visually through allele comparison using Mendelian inheritance. Interestingly, none of the offspring from the bi-parental population HQ2015023, except for USDAHQ2015023.002_FSCR, could have resulted from pollination with the male parent USDA 19048M from the FSCR. Similarly, in the HQ2015034 population none of the offspring were from the male parent USDA 21327M from the FSCR, except for USDA HQ2015034.014_FSCR. This could have been due to a new pollination method that was used to protect flowers on both female parents due to their short arm lengths. Rather than covering individual arms, a large bag was used to cover the axis. The genotype data indicates that the large bags did not sufficiently protect the female flowers, as the alleles in the offspring did not include those in the listed pollen parent.

COLONY did not identify USDA HQ2015034-014 as a potential offspring of the male parent USDA 21327M, although visual inspection of the allele calls of USDA HQ2015034-014 indicated that USDA 21327M could have been the male parent (S13 Table). Similarly, COLONY indicated that Blisk CHUM 166 was most likely the offspring of USDA 21052 representing FSCR's "Atlas", yet visual inspection indicated that Blisk CHUM 166.002 was not an offspring of 'Atlas'. Genepop identified four markers with a moderate to a high frequency of

null alleles (Table 2), including at Primer 6 where 'Blisk' didn't share an allele with its parent 'Atlas', likely due to the presence of a null allele. Null alleles result from lack of PCR amplification [51,116] and pose real challenges for parental reconstruction software and visual inspection because it is difficult to discern homozygous alleles from heterozygous null alleles. Visual inspection of the alleles at the four markers indicated possible heterozygous null allele calls, which could result in parentage conflicts in COLONY and visual inspection. Thus, it is important whenever possible to choose SSRs that do not contain null alleles to prevent conflicting results from parent exclusion [116].

## Identity verification with the 9-SSR and 25 KASP fingerprint sets

The 9-SSR fingerprinting set had a similar distinguishing ability to the 25 KASP assays, except in the WNA samples where the KASP assay did not distinguish unique samples. Hence, a larger number of multi-allele genotypes were generated by the SSR fingerprinting set than that observed in the SNP fingerprinting set in WNA germplasm (Table 1). Low polymorphism and distinguishing ability of this KASP assay was not surprising given that the 25 KASP assays were designed from SNPs selected from *H. l.* var. *lupulus* sequences. In order for the KASP fingerprinting set to distinguish WNA samples, SNPs need to be selected based on polymorphisms in both cultivated and WNA germplasm.

A lower number of SSRs as opposed to SNPs (9 vs. 25, respectively) were required to distinguish the unique *H. lupulus* genotypes. Typically, SSRs are more informative and produce a higher number of alleles at a given locus than SNPs that are bi-allelic [117,118]. Previous studies have shown that a larger number of SNPs is needed to obtain the same information content as in SSR markers [119–122]. However, the ratio of SNPs to SSRs is highly dependent upon the heterozygosity of the samples being studied [121–123]. Singh et al. [123] compared 36 SSRs to 36 SNPs in rice (*Oryza sativa* L.) for 375 samples, including five landraces, 364 modern varieties, and one hybrid. A similar number of alleles was generated for both SSRs and SNPs leading the authors to propose that lower SSR diversity in their study was due to the lower genetic diversity of their samples and higher SSR diversity measures are mostly observed in diverse germplasm. Diversity of the marker also influences the number of markers required in a study. Filipi et al. [121] compared 22 genomic SSR (gSSR) to 20 EST-SSRs and confirmed previous findings that higher levels of diversity are found in gSSR than in EST-SSRs. Similarly, Chen et al. [122] observed that tri-allelic SNPs had higher diversity levels than di-allelic microsatellite markers. Diversity of the marker influences the ratio of SSR to SNP markers because potentially fewer SSR markers would be required if they have a higher information content.

The number of SSRs used in this study to identify European and WNA hop accessions was comparable to other studies. Murakami et al. [36] used 11 SSRs to distinguish 60 wild European, 58 wild Asian, and 15 WNA samples. Bassil et al. [3] used eight genic SSRs to distinguish 24 European and 22 WNA samples. Peredo et al. [18] used seven nuclear SSRs to assess population structure and identify population specific alleles in 68 European, 66 cultivated, and 48 WNA sample.

Previous studies using SNP based genotyping methods have primarily focused on the identification of European and cultivated accessions with limited WNA samples. Matthews et al. [27] and Henning et al. [6] identified 17,128 and 32,206 SNPs using GBS methods to genotype 178 and 121 European and cultivated samples, respectively. Yamauchi et al. [28] identified four SNP rich regions to genotype 21 cultivated hop varieties from Europe and the United States, and Van Holle et al. [45] identified 1,830 highly discriminant SNPs to genotype 56 of the most commercially relevant hop cultivars. KASP markers are starting to gain attention as a valuable genetic tool in plants for genotyping and marker assisted selection (MAS) [124–129].

This is partly because they are a cost-effective option when a low number of markers are needed [125]. KASP markers have even been developed in hop. Jiang et al. [46] developed a 12 KASP marker fingerprinting set to distinguish 16 hop samples. As these markers were developed from only ten of the most commonly used hop varieties in the Chinese brewing industry and were only validated in a very small set of 16 hop samples, we believe the discriminant power of these markers needs to be further evaluated to determine their discriminatory power in closely related individuals and a larger more diverse germplasm set.

In our study, we identified an initial set of 35,633 high quality SNPs from which to select the best minimal subset of GBS markers that differentiated hop accessions consisting of cultivated lines, new experimental breeding lines, and male lines. The minimal set of GBS markers were subsequently developed into KASP markers, of which 25 were ultimately validated on a different set of hop accessions. The 25 KASP markers successfully differentiated cultivated hop accessions but did not perform well at differentiating WNA accessions. Answers concerning this lack of differentiating WNA accessions may be due to WNA accessions containing DNA sequences that are absent in cultivated accessions and therefore not identified in GBS or may be due to the initial set of hop accessions used to develop GBS markers sets not including WNA accessions. Additional GBS work including WNA lines would identify additional KASP markers that could be added to the 25 KASP markers if WNA differentiation was desired. Differentiation among cultivated hops using this set of 25 KASP markers was successful and sufficient for almost all applications.

The 9-SSR fingerprinting set produced a larger number of multi-locus genotypes than the KASP fingerprinting set (for SSR and SNP genotype data, please see data availability). There are two explanations for this. First, in GBS marker identification, all loci having more than two alleles were filtered out due to the expectation that KASP marker development would consist of two alleles. Second, SSR and SNP loci are governed by different mutational processes such as replication slippage and single point mutations, respectively, which lead to different rates of mutation and diversity estimates [130]. The cultivated group had a larger number of multi-locus genotypes and slightly higher diversity statistics in the SSR than in the SNP fingerprinting set. This is likely due to the inability of the KASP SNP fingerprint set to distinguish between WNA samples and European accessions, resulting in placement of a larger number of individuals in the cultivated group. Additionally, SSRs loci have more than two alleles while the chosen SNPs are bi-allelic; thus, it is more likely to find uniqueness between two random samples in a population where there are more possible alleles that can be observed at a locus [86].

## Determining sex at HI-AGA7

Determination of sex at the young seedling stage in hop plants is highly desirable because only the female plants are of economic value and desired for cone production. Poley et al. [131] developed a SCAR marker that was the first sex-specific marker in hop. This marker was found to be associated with the Y chromosome in bulk segregation analysis [40,131]. Later this marker was shown to be in incomplete linkage to the male sex and did not amplify in Japanese males [132,133].

The HI-AGA7 SSR marker was first developed from libraries originating from two hop cultivars Cascade and Savinjski Golding [32]. In 2008, Jakse et al. [7] determined that HI-AGA7 was closely linked to sex in two mapping families grown in Slovenia. McAdam et al. [41] tested HI-AGA7 in a New Zealand mapping population and one of the Slovenian mapping populations [7], concluding that in these two populations an allele at HI-AGA7 was linked to the male sex phenotype. Additionally, HI-AGA7 was used in two previous studies [19,42] because of its high allelic diversity in genotyping, but an association with sex was not examined.

We included HI-AGA7 in our 9-SSR fingerprinting set, which was used to genotype 629 hop samples. When comparing same name accessions with studies of Jakse et al. [7], Korbecka-Glinka et al. [19], and Rodolfi et al. [42], genotypes were comparable with slight variations due to scoring, expect for ten samples. 'Hueller Aroma' from Jakse et al. [7] had a different genotype than that observed in this study and additional investigation is needed to identify a true-to-type 'Hueller Aroma'. Of the nine cultivars that had different genotypes at HI-AGA7 according to Rodolfi et al. [42], eight cultivars had the same genotype in our study and that of Jakse et al. [7], possibly indicating trueness-to-type. The 'El Dorado' sample, however, in this study was obtained from the UNL and its trueness-to-type requires further study.

The high variability at HI-AGA7 was found to be caused by three indels, a variable region after the microsatellite repeat, four transitions, four transversions, and complex variability of the microsatellite repeat itself [7]. Conservation of the flanking sequences allowed primer pairs for this locus to amplify in different species. Such variability is useful in a fingerprinting set and results in high distinguishing ability. However, not all females or males could be identified based on allele composition. Since the sex-specific alleles identified in this study were present in only a limited number of individuals (1 to 6), association with gender needs further validation in additional germplasm.

Recently, a new sex associated marker set was developed by Cerenak et al. [134] by converting four male-specific DArT markers to PCR markers that can be run in a multiplex reaction. These four markers were tested on 197 plants (97 from the world collection and 100 from three segregating families) and showed 100% positive correlation with sex using the multiplex reaction.

## Conclusion

In conclusion, this study provides two cost-effective fingerprinting sets. The first is a set of nine long-core SSR markers that are amplified in a single PCR reaction and able to distinguish cultivated and WNA samples, thus saving time and money. The second is a KASP assay of 25 SNPs that is cost-efficient in distinguishing cultivars in sets of 96 samples each but is unable to distinguish European and WNA accessions. These fingerprinting tools are valuable for identity confirmation and parentage analysis in hop.

## Supporting information

**S1 Fig. Full Radial dendrogram of 629 accessions evaluated with the 9-SSR fingerprinting set.** K-medoids identified three clusters. K-medoids clustering results for each accession are displayed as different colors for each group.
(TIFF)

**S1 Table. List of accessions used in SSR and KASP genotyping.** Samples are listed under each of the three source programs that provided the leaf tissue. All samples were genotyped with the 9-SSR multiplex set. Includes sample name, other name(s), name in dendrogram, local number, PI number, pedigree, accession type [cultivated (CULT) vs. Wild North American (WNA)], and sex.
(XLSX)

**S2 Table. List of long-core repeat SSRs evaluated in the 16 diverse samples making up the Testing Panel (S1 Table).** Includes name, motif, forward primer sequence, reverse primer sequence, expected allele size range, observed allele size range, number of alleles observed, ease of scoring (1 = easy, 0.5 = medium, and 0 = hard), and SSR source citation. Observed allele size range, observed number of alleles, and ease of scoring assessed using the M13 universal

labeling method of Schuelke, 2000.
(XLSX)

**S3 Table. List of 28 KASP assays designed by LGC Biosearch Technologies and evaluated in 190 samples (S1 Table).** Includes name, primer allele sequences (x, y, and common), alleles (x & y), and sequence motif. The targeted SNP is denoted by brackets [] in the sequence.
(XLSX)

**S4 Table. Genotypes for the 9-SSR multiplex for bi-parental populations HQ2015023 and HQ2015034.** Alleles in common with the female parent are highlighted in orange and are not highlighted if shared with the male parent. Alleles that are not shared with either parent are highlighted in grey.
(XLSX)

**S5 Table. Number of SNPs identified after each filtering step to identify the 28 SNPs to develop into KASP.** Filtering steps are listed under each software program and are in order of occurrence.
(XLSX)

**S6 Table. Comparison of the 28 KASP SNP cluster calls to the original genotyping-by-sequence (GBS) bi-allelic SNP calls in sixty-three accessions that had the same name.** The three excluded KASP assays are highlighted in orange and are excluded from analysis. Individuals that had different genotypes between KASP and GBS at a locus are highlighted in grey. Failed KASP assays are highlighted in red and are excluded from analysis.
(XLSX)

**S7 Table. Summary comparison of allele calls and allele mismatch type in 63 individuals with the same name genotyped by GBS and KASP at the same 28 loci.** Summary means and totals were calculated in all 63 individuals after excluding samples with >20% mismatched allele calls (highlighted in grey) between GBS and KASP. Samples that have homozygous GBS calls that were homozygous for the opposite allele in KASP are highlighted in orange.
(XLSX)

**S8 Table. Summary comparison of GBS to KASP mismatch type at the 28 loci.** Highlighted markers were excluded from analysis because they contained >20% missing data and/or didn't form three distinct clusters corresponding to the two homozygous and one heterozygous genotype.
(XLSX)

**S9 Table. Eighty-nine synonym sets that had the same fingerprints and were given a group name to represent the corresponding genotype.** Grey samples are contamination, propagation, or source errors.
(XLSX)

**S10 Table. COLONY full likelihood (FL) and pairwise likelihood (PL) parentage analysis.** Includes offspring, candidate father, father FL probability, father PL confidence level, candidate mother, mother FL probability, mother PL confidence level, and parent PL probability. Parent PL probability does not include candidate male if father FL probability is < 0.50. FL probability and PL probabilities range from 0 to 1. Higher FL probability and PL level of confidence numbers indicate a stronger parental candidate. Higher parent PL probability indicates stronger parental pair configuration. Missing values (-) indicate COLONY didn't identify a candidate parent. The likely true to type genotype is highlighted in grey.
(XLSX)

**S11 Table. COLONY best maximum likelihood (ML) full family configuration.** Each full sibling relationship is supported by full likelihood > 0.50 probability. Includes mother, father, probability of inclusion (Inc.), probability of exclusion (Exc.), and siblings included in the full sib family. Probability of inclusion ranges from 0 to 1 with low numbers indicating family is splitable. Probability of exclusion ranges from 0 to probability of inclusion of the same full sib family with higher numbers indicating that individuals within a full sib configuration are full sib and no other individuals are full sibs within the full sib family. Higher probability of inclusion and lower probability of exclusion means that the best full sibship is probably true, but incomplete.
(XLSX)

**S12 Table. Genotype of same name accessions from the USDA-ARS NCGR and the USDA-ARS FSCR that had different profiles.** Includes female parent, cultivar set name, sample names, and the genotype at each SSR in the 9-SSR fingerprinting set. Alleles that are different within each cultivar group are highlighted in grey.
(XLSX)

**S13 Table. Genotypes at the 9 SSRs for female parents and nine reported progenies with non-matching allele composition that were obtained from the USDA-ARS NCGR and the USDA-ARS FSCR.** Samples are grouped by common female parent. Alleles in common with the female parent are highlighted in orange. Offspring alleles that are not shared with the female parent are highlighted in grey.
(XLSX)

**S14 Table. Eighteen synonym sets and *H. lupulus* var. *neomexicanus* Colorado 2–1 of 190 samples genotyped with the 9-SSR and the 25-KASP assays.** Synonym sets were given a group name to represent the corresponding genotype. Includes synonym sets identified by genotyping 190 samples with the 9-SSR set collected independently from those submitted to LGC Biosearch Technologies DNA for KASP assay genotyping, 9-SSR synonym sets after allele corrections with DNA from LGC Biosearch Technologies.
(XLSX)

**S15 Table. Comparison of HI-AGA7 genotype of hop cultivars from this study that are in common with three previous studies: Jakse et al., 2008; Korbecka-Glinka et al., 2016; and Rodolfi et al., 2018.** Includes, cultivar name, hop type [cultivated (CULT) vs wild North American (WNA)], sex, HI-AGA genotype for accessions in each of the studies. Samples highlighted in gray do not have equivalent allele calls. (-) indicates no sample or genotype.
(XLSX)

**S16 Table. Equivalent alleles at HI-AGA7 between this study and three previous studies: Jakse et al., 2008; Korbecka-Glinka et al., 2016; and Rodolfi et al., 2018.** (-) indicates inability to identify equivalent allele due to absence of genotypes in common between our study and the study in question.
(XLSX)

**S17 Table. List of 30 alleles identified at HI-AGA7 alleles per sex in cultivated (CULT), wild North American (WNA) or both (BOTH) types of hop germplasm.** Allele counts in females and males are listed. Alleles highlighted in light grey are male specific and in dark grey are female specific. (-) indicates allele was not found.
(XLSX)

## Acknowledgments

We appreciate the assistance of April Nyberg and Jaimie Green for technical support and that of Jeanine DeNoma, Jim Oliphant, Debra Hawkes, and Gabriel Flores for management of the hop germplasm collection at NCGR. We thank Jernej Jakse at the Agronomy Department, Biotechnical Faculty, University of Ljubljana, Ljubljana, Slovenia for providing the allele data for HI-AGA7.

## Author Contributions

**Conceptualization:** Keenan Amundsen, Annette Wiles, Claudia Wiedow, Josef Patzak, John A. Henning, Nahla V. Bassil.

**Data curation:** Mandie Driskill, Nahla V. Bassil.

**Formal analysis:** Nahla V. Bassil.

**Funding acquisition:** Keenan Amundsen, Annette Wiles, Nahla V. Bassil.

**Investigation:** Nahla V. Bassil.

**Methodology:** Katie Pardee, Kim E. Hummer, Jason D. Zurn, Nahla V. Bassil.

**Project administration:** Nahla V. Bassil.

**Resources:** Kim E. Hummer, John A. Henning, Nahla V. Bassil.

**Supervision:** Nahla V. Bassil.

**Writing – original draft:** Mandie Driskill, Nahla V. Bassil.

**Writing – review & editing:** Mandie Driskill, Nahla V. Bassil.

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
