## [Decision Letter · Decision Letter 0]

12 Nov 2021

PONE-D-21-27320Two Fingerprinting Sets for Humulus lupulus Based on KASP and Microsatellite MarkersPLOS ONE

Dear Dr. Bassil,

Thank you for submitting your manuscript to PLOS ONE. After careful consideration, we feel that it has merit but does not fully meet PLOS ONE’s publication criteria as it currently stands. Therefore, we invite you to submit a revised version of the manuscript that addresses the points raised during the review process.

We look forward to receiving your revised manuscript.

Kind regards,

Vikas Sharma, Ph.D

Academic Editor

PLOS ONE

Additional Editor Comments (if provided):

This research work provided important sets of KASP and SSR markers for Humulus lupulus which will be useful for its characterization and authentication in future works. Overall, manuscript is well written and every related aspect has been well documented and presented. I strongly recommend its publication in PLOS ONE after careful revision to remove lingual errors etc. (line no. 191, line 207: OR can be written in full, line no. 232: using used, line no. 269: sentence starting from sNMF) please correct all such minor issues throughout the manuscript before publication.

Journal Requirements:

“We thank USDA CRIS 2072-21000-049-00D and CRIS 2072-21000-051-00D for financial support for this project. We thank the Brewers Association for additional funding of this project.”

“We thank USDA CRIS 2072-21000-049-00D and CRIS 2072-21000-051-00D for financial support for this roject. We thank the Brewers Association for additional funding of this project.”

We note that you have provided additional information within the Acknowledgements Section. Please note that funding information should not appear in the Acknowledgments section or other areas of your manuscript. We will only publish funding information present in the Funding Statement section of the online submission form.

“We thank USDA CRIS 2072-21000-049-00D and CRIS 2072-21000-051-00D for financial support for this project. We thank the Brewers Association for additional funding of this project.”

4. Your abstract cannot contain citations. Please only include citations in the body text of the manuscript, and ensure that they remain in ascending numerical order on first mention.

Reviewers' comments:

Reviewer's Responses to Questions

**Comments to the Author**

1. Is the manuscript technically sound, and do the data support the conclusions?

Reviewer #1: Yes

2. Has the statistical analysis been performed appropriately and rigorously? 

Reviewer #1: Yes

3. Have the authors made all data underlying the findings in their manuscript fully available?

Reviewer #1: Yes

4. Is the manuscript presented in an intelligible fashion and written in standard English?

Reviewer #1: Yes

5. Review Comments to the Author

Reviewer #1: The manuscript is in good shape and properly crafted. The problem and objectives are identified in a systematic way. The methodology of the experiment is scientific, classified and well designed. The dataset and sample pool is reasonably justifiable. The manuscript is written in good, lucid, simple and grammatically correct language. Statistical tools used for the analysis are robust and latest with updated references. The manuscript is recommended for the publications after checking the requirements and formats of the journal.

6. PLOS authors have the option to publish the peer review history of their article (what does this mean?). If published, this will include your full peer review and any attached files.

---

## [Author Response · Author response to Decision Letter 0]

11 Feb 2022

We have done the following:

1. We have removed the financial disclosure from the acknowledgement section

2. We would like the Financial Disclosure to be:

‘We thank USDA CRIS 2072-21000-049-00D and CRIS 2072-21000-051-00D for financial support for this project. We thank the Brewers Association for additional funding of this project. The funders had no role in study design, data collection and analysis, decision to publish, or preparation of the manuscript.’

3. There were no citations in the abstract

4. We deleted “data not shown” in our manuscript since the data we referred to is not a core part of the research being presented.

5. All data for this article is publicly available and was stated clearly in Data Availability

6. We removed lingual errors etc. across the manuscript as you see in the ‘Manuscript with Track Changes’

7. We checked the format requirements at https://journals.plos.org/plosone/s/file?id=wjVg/PLOSOne_formatting_sample_main_body.pdf and changed the format to meet it

---

## [Editor Report · Decision Letter 1]

7 Mar 2022

Two Fingerprinting Sets for Humulus lupulus Based on KASP and Microsatellite Markers

PONE-D-21-27320R1

Dear Dr. Bassil,

We’re pleased to inform you that your manuscript has been judged scientifically suitable for publication and will be formally accepted for publication once it meets all outstanding technical requirements.

Kind regards,

Vikas Sharma, Ph.D

Academic Editor

PLOS ONE

Additional Editor Comments (optional):

Authors have revised the manuscript as per minor comments raised previously. Therefore, it can be accepted now. However, authors are advised to revise manuscript to remove any minor errors (lingual or any other) as afterwards no changes will be possible in the manuscript.
---

## [Editor Report · Acceptance letter]

5 Apr 2022

PONE-D-21-27320R1 

Two Fingerprinting Sets for *Humulus lupulus* Based on KASP and Microsatellite Markers 

Dear Dr. Bassil:

I'm pleased to inform you that your manuscript has been deemed suitable for publication in PLOS ONE. Congratulations! Your manuscript is now with our production department. 

Kind regards, 

on behalf of

Dr. Vikas Sharma 

Academic Editor

PLOS ONE